# Microglial REV-ERBα regulates inflammation and lipid droplet formation to drive tauopathy in male mice

Jiyeon Lee [1], Julie M. Dimitry[1], Jong Hee Song [2], Minsoo Son[2], Patrick W. Sheehan [1], Melvin W. King[1], G. Travis Tabor[3], Young Ah Goo[2], Mitchell A. Lazar [4], Leonard Petrucelli[5] & Erik S. Musiek [1] ✉

Alzheimer's disease, the most common age-related neurodegenerative disease, is characterized by tau aggregation and associated with disrupted circadian rhythms and dampened clock gene expression. REV-ERBα is a core circadian clock protein which also serves as a nuclear receptor and transcriptional repressor involved in lipid metabolism and macrophage function. Global REV-ERBα deletion has been shown to promote microglial activation and mitigate amyloid plaque formation. However, the cell-autonomous effects of microglial REV-ERBα in healthy brain and in tauopathy are unexplored. Here, we show that microglial REV-ERBα deletion enhances inflammatory signaling, disrupts lipid metabolism, and causes lipid droplet (LD) accumulation specifically in male microglia. These events impair microglial tau phagocytosis, which can be partially rescued by blockage of LD formation. In vivo, microglial REV-ERBα deletion exacerbates tau aggregation and neuroinflammation in two mouse tauopathy models, specifically in male mice. These data demonstrate the importance of microglial lipid droplets in tau accumulation and reveal REV-ERBα as a therapeutically accessible, sex-dependent regulator of microglial inflammatory signaling, lipid metabolism, and tauopathy.

REV-ERBα is a circadian clock protein and transcriptional repressor which has pivotal roles in diverse physiological processes such as inflammation, lipid/glucose metabolism, and circadian rhythm[1]. REV-ERBα is a nuclear receptor that exhibits strong circadian rhythms in expression in the brain, and we previously found that global germ-line deletion of REV-ERBα in mice evoked glial activation and neuroinflammation in the brain[2]. In general, glia cells can clear debris, aggregated proteins, and damaged neurons for the maintenance of brain health[3]. Microglia in particular are crucial mediators of inflammation, and our previous studies have revealed that global REV-ERBα can increase microglia phagocytic activity and reduce Aβ deposition in

5xFAD mouse brain[4]. Thus, REV-ERBα has potential therapeutic benefits in neurodegenerative diseases including Alzheimer's disease (AD) through modulation of glial function. However, the cell-specific functions of REV-ERBα in microglia and other glial cells are unknown.

Although REV-ERBα is known as a mediator of lipid metabolism in periphery, the exact role of REV-ERBα in lipid metabolism in the central nervous system (CNS), especially in microglia, is unclear. Recent studies show that lipid homeostasis and lipid droplet (LD) formation are disrupted in microglia in AD models and LD formation in microglial has been highlighted as a potential therapeutic target[5–7]. Lipid-droplet accumulating microglia (LDAM) have been observed in mouse and

[1]Department of Neurology and Center On Biological Rhythms And Sleep, Washington University School of Medicine, St. Louis, MO, USA. [2]Mass Spectrometry Technology Access Center at McDonnell Genome Institute (MTAC@MGI) at Washington University School of Medicine, St. Louis, MO, USA. [3]Department of Neurology, Hope Center for Neurological Disorders, Knight Alzheimer's Disease Research Center, Washington University School of Medicine, St. Louis, MO, USA. [4]Institute for Diabetes, Obesity, and Metabolism, Perelman School of Medicine, University of Pennsylvania, Philadelphia, PA, USA. [5]Department of Neuroscience, Mayo Clinic, Jacksonville, FL, USA. ✉e-mail: musiek@wustl.edu

human brain during aging and neurodegenerative diseases and are an emerging pathological phenotype in AD[8]. Perilipin 2 (PLIN2) has been used as a representative surface marker of LDs[9] and is increased in microglia after lipopolysaccharide (LPS) treatment[10] and in neurodegenerative conditions[8]. However, the function of LDs in microglia, particularly in the setting of AD, is poorly understood.

Microglial activation is considered one of the hallmarks of neuroinflammation and is observed in AD and other neurodegenerative disease[11,12]. Microglia respond to brain pathology, and single-cell RNA-sequencing (scRNA-seq) has revealed various clusters of microglia that have been mapped to activated and homeostatic states under resting and pathological conditions[13,14]. Recent transcriptomic studies have revealed striking sex differences in microglial gene expression and function, as male microglia appear more activated compared to female microglia under basal conditions and exert stronger pro-inflammatory responses to stimuli such as stress and pathogenic insults[15,16]. Strikingly, ischemic damage in male mice can be rescued by transplantation of female microglia[16]. Diverse neurological disorders including AD show strong sex differences, suggesting that the impact of sex on microglial function might influence disease risk.

Here, we propose a role for the circadian protein, REV-ERBα, in regulating inflammatory tone and LD expression in microglia. We find that REV-ERBα-deficient microglia exhibit increased inflammatory signaling, LD accumulation, and impaired tau phagocytic activity. In vivo, microglia-specific REV-ERBα deletion exacerbated neuroinflammation and tau pathology in two mouse models. These effects of REV-ERBα deletion on microglial LD formation, tau phagocytosis, inflammation, and eventually tauopathy were only observed in male cells and mice, suggesting that REV-ERBα regulates sex-dependent responses under tau-related pathological conditions. More broadly, we find that inflammation and LD formation interact in microglia to impair tau phagocytosis, and that exposure to tau-enriched brain extract can trigger these pathways. Our studies highlight the importance of microglial LDs in AD pathogenesis and identify REV-ERBα as a regulator of these processes and potential therapeutic target.

## Results

### Validation of microglial REV-ERBα KO mouse model and its impact on neuroinflammation

According to our previous report, global germ-line REV-ERBα deletion causes general neuroinflammation and affects microglia activation state[2,4]. To better understand the specific role of microglial REV-ERBα in vivo, we generated tamoxifen-inducible, microglia-specific REV-ERBα knockout (KO) mice (Cx3cr1::Cre[ERT2]; Nr1d1[fl/fl]) (Fig. 1a). This Cre line allows specific targeting of microglia, as non-microglial Cx3cr1+ myeloid cells repopulate 6−8 weeks after tamoxifen treatment[17]. We observed specific deletion of exon 3−5 of REV-ERBα (Nr1d1) and >-90% deletion of Nr1d1 transcript in isolated microglia from our tamoxifen-treated Cre+ mice, demonstrating efficient Cre−mediated recombination (Fig. 1b, c).

Using this model, we next measured neuroinflammation by parameters such as glial activation and inflammatory cytokine expression. Unlike global germ-line REV-ERBα KO mice[2], we did not observe spontaneous microglial activation, as assessed by staining for microglial markers including IBA1 and CD68, though GFAP-positive astrocytes were modestly increased in microglial REV-ERBα KO hippocampus (Fig. 1d, e). This suggests that REV-ERBα-deficient microglia may exert some mild increased inflammatory tone which is not sufficient to induce the severe neuroinflammatory phenotype described in germ-line REV-ERBα KO. Although microglial REV-ERBα KO showed only a mild effect on glial activation in vivo, inflammatory cytokine transcripts such as Il1b and Il6 were significantly increased in cultured Cre+ microglia from microglial REV-ERBα KO pups treated with 4-hydroxy-tamoxifen in vitro, as compared to control (Cre−) cells (Fig. 1f). These data suggest that microglial REV-ERBα deletion causes modest basal inflammation in the brain.

### Microglial REV-ERBα KO exacerbates tau pathology and neuroinflammation specifically in male PS19 mice

To investigate the role of microglia REV-ERBα on AD-related pathology in vivo particularly tauopathy, we crossed microglia-specific REV-ERBα KO mice with the P301S PS19 human tau transgenic mouse line[18], to generate Cx3cr1::Cre[+/−]; Nr1d1[fl/fl]; P301S[+/−] mice and Cre− littermate controls. All mice were treated with tamoxifen at 8 weeks of age to induce REV-ERBα deletion (in Cre+ mice), and harvested at 10 months old, when male mice have moderate tau pathology (Fig. 2a). As shown in Fig. 2b, c, microglial REV-ERBα KO significantly increased the levels of AT8-positive phosphorylated tau (pTau) in hippocampus and cortex of male mice. Transcripts for pro-inflammatory cytokines such as Il1b and Tnfa, complement component C1q, and activated microglia marker Cd68 were also significantly induced in hippocampus of Cre+ PS19 male mice, as compared to PS19; Cre− mice (Fig. 2d). Moreover, immunostaining for activated astrocytes (GFAP) and microglia (IBA1/CD68) showed increased tau-related gliosis in Cre+ mice (Fig. 2f). However, we observed that, unlike males, female microglia-specific REV-ERBα KO mice showed no increase in pTau (AT8) pathology (Fig. 2b), pro-inflammatory cytokines (Fig. 2e), or glial activation (Fig. 2f) compared to PS19; Cre− female controls. We also measured the thickness of the neuronal cell body layer of hippocampal CA1 and dentate gyrus using NeuN staining, as a measure of neurodegeneration[19]. We observed that CA1 thickness was slightly but significantly decreased in male Cre+ PS19 mice (Fig. S1a), but not female (Fig. S1b), in keeping with our findings of increased tau pathology in male Cre+ mice. Thus, the effects of microglia REV-ERBα on tau pathology and subsequent inflammation in PS19 mice are highly sex-dependent and result in exacerbated pathology only in male mice.

### Microglial REV-ERBα KO exacerbates pTau accumulation and gliosis in male mice in an AAV-P301L tau model

In general, female PS19 mice have slower pTau deposition during aging than male mice[20]. Thus, we next sought to confirm our findings from the PS19 genetic model in a second tauopathy model with similar levels of pathology between sexes. We chose an AAV-P301L viral tauopathy model, which has been shown to induce consistent and widespread aggregated tau pathology within 6 months[21]. We performed intracerebroventricular (i.c.v) injection of AAV-P301L, into Cx3cr1::Cre[ERT2]; Nr1d1[fl/fl] pups (and Cre− littermate controls) at post-natal day 0 and gave tamoxifen 2 months later to delete REV-ERBα in Cre+ microglia (Fig. 3a). At 6 months old, we measured tauopathy by staining for pTau with AT8 and for misfolded tau with MC1 antibodies. We also stained total human tau expression using HT7 antibody and used this to normalize AT8 and MC1 expression levels, to account for variation in viral vector expression. As in the tau PS19 mice, hippocampal AT8+ and MC1+ cells were significantly increased in male microglial REV-ERBα KO mice compared to Cre− controls (Fig. 3b, d), but not in females (Fig. 3c, e). Astrocyte (GFAP) and microglial activation (IBA1 and CD68) were also increased in parallel with tau pathology in male Cre+ mice whereas both were decreased in female Cre+ mice. (Fig. 3f, g). Altogether, microglial REV-ERBα KO consistently increases tau pathology and gliosis across tauopathy models in male mice but slightly reduces these endpoints in female mice. Thus, the effects of microglia REV-ERBα are highly sex-dependent and result in differential effects on tau pathology in male and female mice.

### LPS-mediated inflammation is also amplified by microglial REV-ERBα KO in male mice

To examine the sex-dependent inflammatory responses of microglia[15,16], we generated primary cultured microglia from individual pups and genotyped for sex using markers of X and Y chromosome genes, Myog and Sry[22,23], respectively (Fig. S2a, S2b). As previous reports have shown that cultured astrocytes have differential responses by LPS by sex[24,25], we tested whether cultured microglial also have

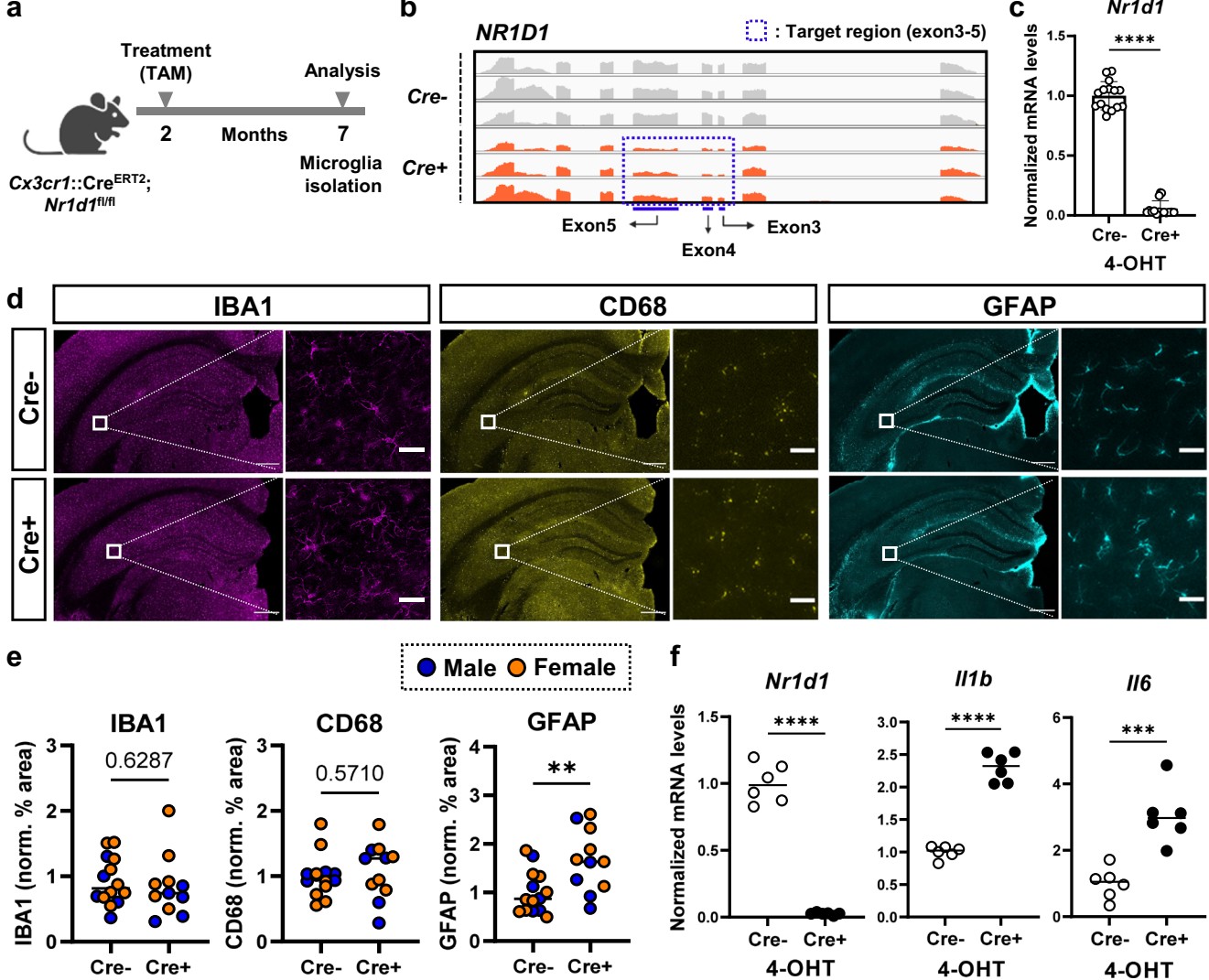

**Fig. 1 | Microglial REV-ERBα KO causes neuroinflammation in vivo and in vitro.**
**a** Schematic showing experiments with microglia-specific REV-ERBα KO mice
(*Cx3cr1*::Cre$^{ERT2}$;*Nr1d1*$^{fl/fl}$ mice). Created with Biorender.com. **b** Integrative genomics
viewer (IGV) snapshot of RNA-seq reads at the floxed REV-ERBα locus (Exon 3, 4,
and 5) of isolated microglia from tamoxifen-injected *Cx3cr1*::Cre$^{ERT2}$;*Nr1d1*$^{fl/fl}$ mice
and Cre− littermate controls. **c** Rev-erbα (*Nr1d1*) KO efficiency in isolated microglia
from *Cx3cr1*::Cre$^{ERT2}$; *Nr1d1*$^{fl/fl}$ pups after 4-OHT treatment (1.5 μM) (*n* = 15 wells
examined from 3 cultures derived from independent pups). **d**, **e** Astrocyte (cyan:

GFAP) and microglial (purple: IBA1, yellow: CD68) activation in Cre− controls and
Cre+ *Cx3cr1*::Cre$^{ERT2}$; *Nr1d1*$^{fl/fl}$ mouse hippocampus (*n* = 5–9 mice per group). Scale
bar, 500 μm (whole hippocampus); Scale bar, 50 μm (Zoom image). **f** mRNA
expression of pro-inflammatory cytokines such as *Il1b*, and *Il6* in control (Cre−;
*Nr1d1*$^{fl/fl}$) and REV-ERBα KO (*Cx3cr1*::Cre$^{ERT2}$;*Nr1d1*$^{fl/fl}$) microglia from P1-3 pups
treated with 4-OH-TAM (*n* = 6 independent cell cultures). **p* < 0.01, ***p* < 0.005,
*****p* < 0.001, by two-tailed *t* test. *P* values > 0.05 are listed. Error bars
represent SEM.

differential responses to neuroinflammatory stimulation with LPS. We
treated male and female microglial cultures with vehicle (VEH) or LPS
(100 ng) for 24 hrs and measured transcript levels of several cytokines
such as *Il6* and *Il1b*. Notably, male microglia exhibited higher inflam-
matory gene expression in response to LPS than female microglia
(Fig. S2c).

We next compared the responses of male and female
*Cx3cr1*::Cre$^{ERT2}$; *Nr1d1*$^{fl/fl}$ mice and Cre− controls to LPS-induced brain
inflammation. Interestingly, eight hours after a 2 mg/kg intraperitoneal
(i.p) LPS injection (Fig. S2d), we observed greater induction of *Il6* and
*Il1b* transcripts in cerebral cortex tissue from male Cre+ mice, with no
exacerbation of LPS-induced neuroinflammation in female microglial
REV-ERBα KO mice (Fig. S2e). However, gliosis, determined by the
expression of *Gfap* and *Cd68* transcript, tends to already be saturated
toward activation (Fig. S2e). Thus, the pro-inflammatory effects of
microglial REV-ERBα KO only appear upon immune stimulation and
are sex-dependent and restricted to male mice.

## REV-ERBα KO causes abnormal lipid metabolism and induces lipid-droplet accumulation in microglia

We next examined transcriptional changes caused by REV-ERBα dele-
tion in microglia. We cultured control (Cre−) and REV-ERBα KO (Cre+)
microglia from CAG::Cre$^{ERT2}$; *Nr1d1*$^{fl/fl}$ pups, treated with 4-hydroxy-
tamoxifen (4-OHT) for 2 days, and analyzed them by bulk RNA-
sequencing (Fig. 4a). Cluster analysis showed a total of 3750 sig-
nificantly differentially expressed genes (DEG), and 'cell activation' and
'cellular response to interferon-gamma' were identified by GO term
pathway analysis as the top functions that become dysregulated in
REV-ERBα KO cells (Fig. 4b). Aside from inflammatory-related path-
ways, "Lipid metabolic process" also drew our attention, as the lipid
regulating functions of REV-ERBα in the periphery are well known[26].
Numerous lipid metabolic transcripts were either up- or down-
regulated in Cre+ microglia, including several genes involved with
glutathione metabolism (*Gsta3, Gstm1, Gstp1, Mgst2*), sphingomyelin
synthesis (*Sgms2*), and lipid transport (*Apoe*) (Fig. 4c).

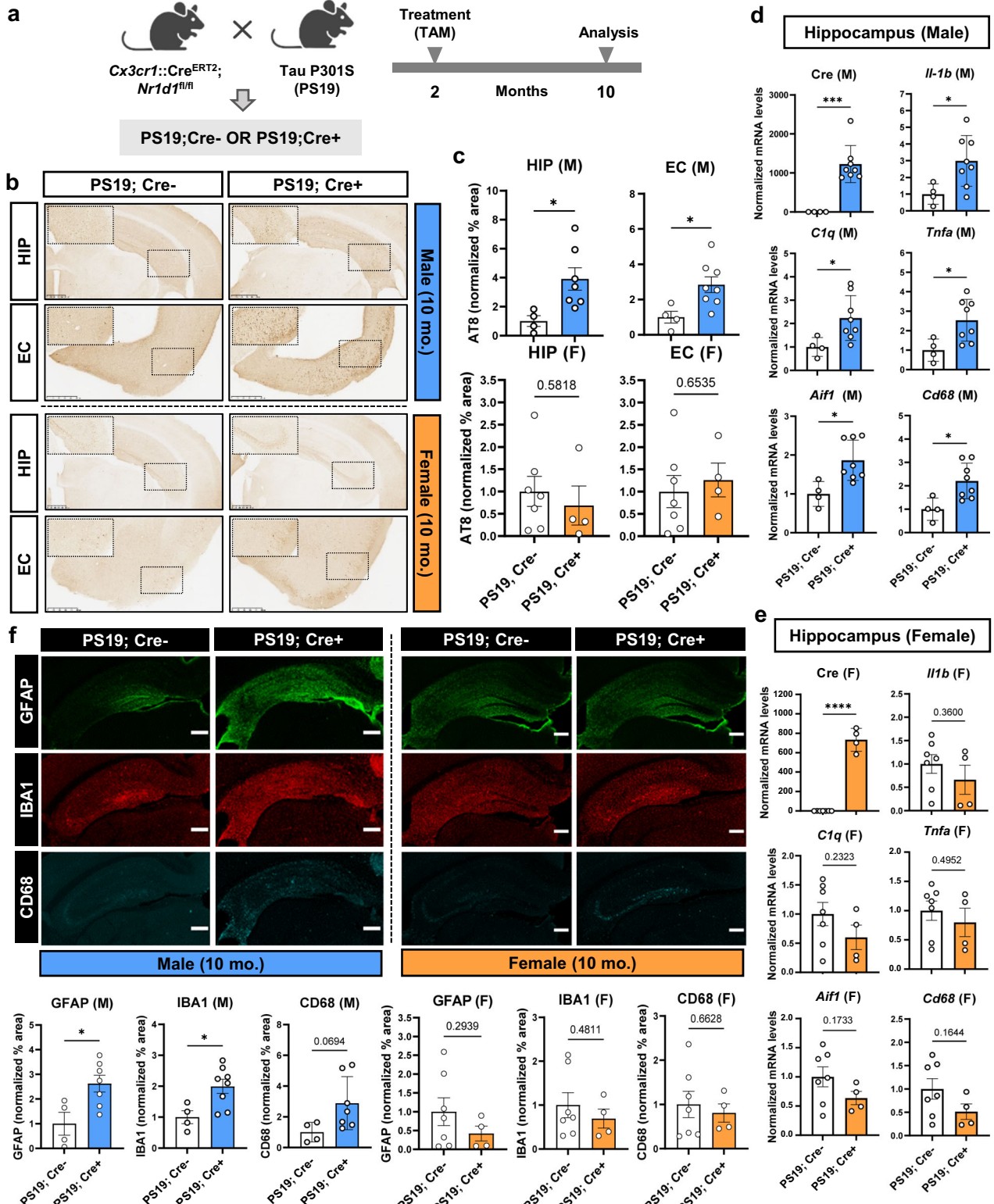

**Fig. 2 | Microglial REV-ERBα KO accelerates pTau accumulation and neuroinflammation in PS19 male mice only. a** Schematic for generating P301S tau-expressing microglial REV-ERBα KO mice (PS19⁺/⁻;Cx3cr1::Creᴱᴿᵀ²⁺;Nr1d1ᶠˡ/ᶠˡ). Created with Biorender.com. **b** Hippocampal and cortical pTau staining with AT8 in control P301S (PS19; Cre−) and microglial REV-ERBα KO P301S (PS19; Cre+) male and female mice, quantified in **c** (n = 4–8 mice per group). Hatched box highlights CA1 and cortical region where differences in AT8 were most apparent. **d** Expression of

transcripts for pro-inflammatory cytokines (Il1b, Tnfα), complement component C1q, and microglia markers (Aif1, Cd68) in the hippocampus of PS19; Cre− and PS19; Cre+ male (n = 4–8 mice per group) and **e** female mice (n = 4–7 mice). **f**, Immunostaining for astrocytes (GFAP; green) and microglia (IBA1; red, CD68; cyan) in male and female mice, with quantified percentage area normalized to Cre− group (n = 4–8 mice per group). Scale bar, 500 μm. *p < 0.05 and ***p < 0.005, ****p < 0.001 by two-tailed t test. P values > 0.05 are listed. Error bars represent SEM.

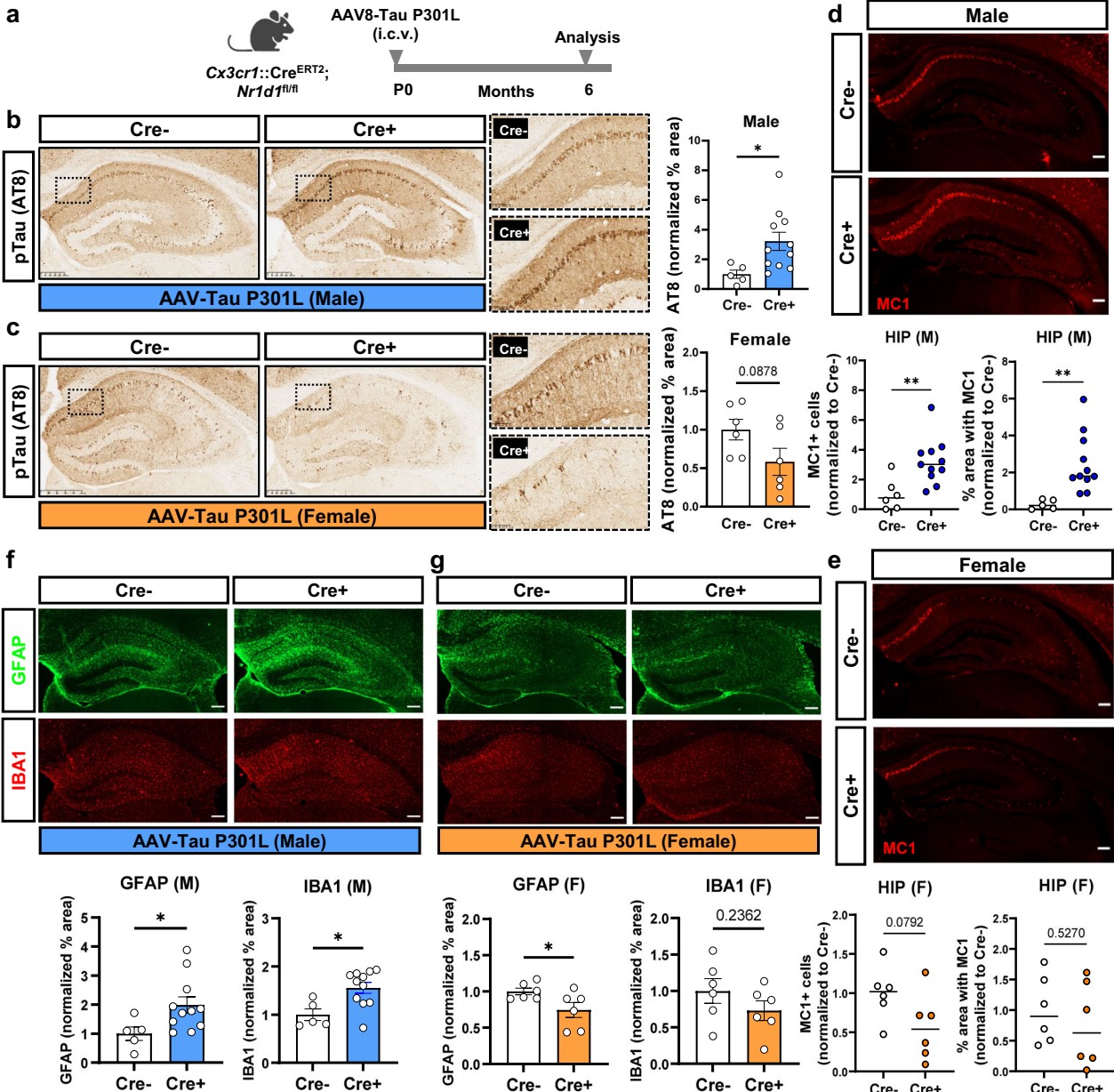

**Fig. 3 | Microglial REV-ERBα KO exacerbates tauopathy in males only in an AAV-tau P301L model. a** Schematic depicting post-natal P0 intracerebroventricular (i.c.v.) injection of AAV-tau P301L into Cx3cr1::Cre^ERT2+;Nr1d1^fl/fl mice and Cre− controls. Created with Biorender.com. **b** Images and quantification of AT8 immunoreactivity in hippocampus of male (*n* = 5–11 mice per group) and **c** female Cre− control and Cre+ microglial REV-ERBα KO mice 6 months after AAV-tau P301L injection (*n* = 6 mice per group). AT8 intensity normalized to total tau staining (HT7 antibody) for each mouse, then the percentage area of AT8 staining was normalized

to the Cre− group. Hatched box highlights CA1 region where differences in AT8 were most apparent. **d** MC1 immunoreactivity in hippocampus of control (Cre−) and microglial REV-ERBα KO (Cre+) AAV-tau P301L expressing male (*n* = 5–11 mice per group) and **e** female mice (*n* = 6 mice per group). Percent area was normalized to Cre− group. **f** Immunostaining of astrocytes (GFAP; green) and microglia (IBA1; red) in hippocampus of Cre− or Cre+ male (*n* = 5–11 mice per group) and **g** female AAV-P301L injecting mice (*n* = 6 mice per group). Scale bar, 500 μm. *p < 0.05 by two-tailed *t* test. P values > 0.05 are listed. Error bars represent SEM.

Altered lipid metabolism and lipid droplet accumulation in inflammatory macrophages and microglia appear to influence cellular function[5,27,28], and REV-ERBα KO microglia exhibit transcriptional changes consistent with increased inflammation and altered lipid metabolism. For this reason, we next investigated if these two processes could be interrelated. We observed that 24-h treatment of cultured microglia with LPS dramatically increased inflammatory cytokine expression (Fig. 4d) and also caused increased accumulation of lipid droplets (LDs), as assessed using the neutral lipid marker BODIPY 493/503 by flow cytometry (Fig. 4e) with sequential gating strategies (Supplementary Fig. S3a). Accordingly, REV-ERBα KO microglia showed dramatically increased levels of LDs under basal conditions, as quantified by BODIPY

staining (Fig. 4f–h) and both male and female REV-ERBα KO microglia showed increased LD accumulation. Moreover, *Plin2*, an LD marker, was also significantly increased in siRev-erbα transfected cultured microglia compared to siControl group (Fig. 4i). Taken together, these data suggest that REV-ERBα KO in microglia was accompanied by transcriptomic changes in lipid metabolic pathways and LD accumulation, though these were not sex-dependent under basal conditions.

## LDs accumulate in microglia in human AD and aged REV-ERBα KO mice
Perilipins (PLINs) are surface markers of LDs under normal physiological conditions[9]. To assess the involvement of LDs in AD, we

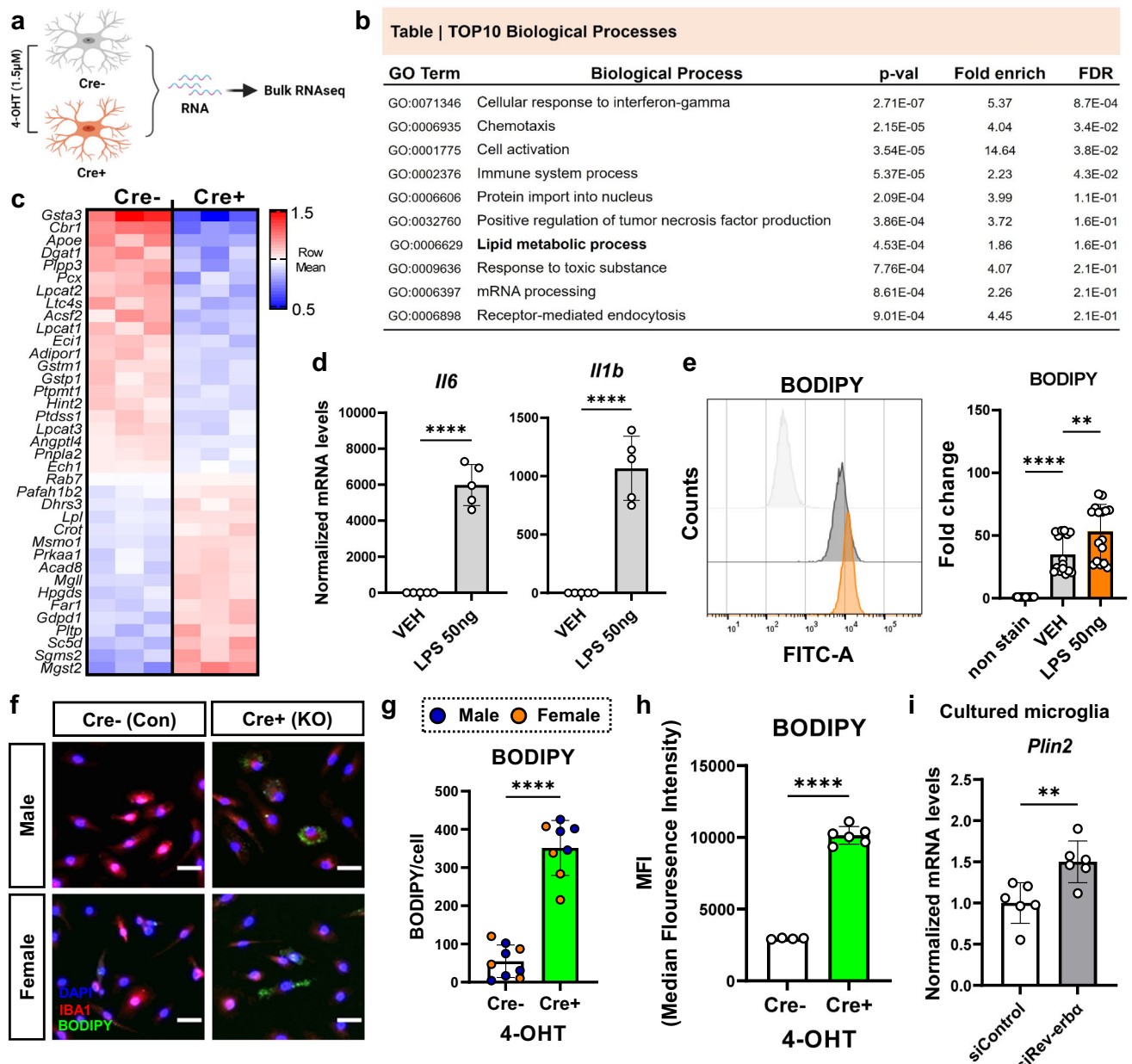

**Fig. 4 | REV-ERBα KO alters lipid metabolism and the expression of lipid-droplet in microglia. a** Schematic illustrating 4-OHT treated Cre− control and Cre+ REV-ERBα KO (RKO) microglia for bulk RNA-seq. Created with Biorender.com. **b** Selected TOP 10 biological processes identified for DEGs in REV-ERBα KO microglia from bulk RNA-seq. **c** Heat-map representing genes from "Lipid metabolic process" GO term. Red: upregulation; Blue: downregulation. **d** Responses of inflammatory genes including *Il6* and *Il1b* to 50 ng/ml LPS exposure in WT cultured microglia (*n* = 5 biologically independent samples). **e** Lipid-droplet staining using BODIPY 493/503 dye in 50 ng/ml LPS-treated WT cultured microglia by flow cytometry (*n* = 14 examined over four independent experiments). **f** Representative images of BODIPY+ signal in 4-OHT treated male/female Cre− control or Cre+ RKO cultured microglia (green; BODIPY, red; IBA1, blue; DAPI) and **g**, quantified BODIPY + signals per cell (*n* = 8–9 independent cell cultures). Scale bar, 100 μm. **h** Mean Fluorescence Intensity (MFI) of BODIPY+ cells in 4-OHT-treated Cre− control and Cre+ RKO microglia using flow cytometry (*n* = 4–6 independent cell cultures). **i** Increased Plin2 expression in microglia transfected with control (siControl) or Rev-erbα (siRev-erbα) siRNA (*n* = cells from 6 pups/genotype). \*\**p* < 0.01, \*\*\**p* < 0.005, \*\*\*\**p* < 0.001, by two-tailed *T* test or two-way ANOVA with Sidak multiple comparisons test. *P* values > 0.05 are listed. Error bars represent SEM.

analyzed the levels of all five different PLIN family member transcripts (*PLIN1* to 5) using an existing microarray dataset (GSE5281) from human AD brain[29]. Human *Plin2* and *Plin3* were highly expressed in entorhinal cortex (EC) tissue from AD patient samples as compared to controls (Fig. 5a). In the same AD transcriptomics dataset, we observed decreased *Nr1d1* (REV-ERBα) expression in AD patient samples (Fig. 5a), suggesting that diminished REV-ERBα could contribute to abnormal high expression of *PLIN2* (and perhaps increased LDs) in AD brain. Moreover, single nucleus RNA-seq data from the Seattle Alzheimer's Disease Brain Cell Atlas (SEA-AD)

showed enriched levels of PLIN2 in the microglia cluster, particularly among 24 individual clusters based on the cell types in AD brain (Fig. 5b, c). Of note, *Plin2* expression was originally described as a transcript defining the disease-associated microglia (DAM) phenotype in 5xFAD mice[30] and isolated microglia from tau transgenic mice (PS19) also show *Plin2* as a DEGs compared to WT mice[5,30–32]. Based on these transcriptomic databases, we considered microglia as the primary site of *Plin2* expression in the brain in the setting of AD pathology, illustrating potential relevance to AD pathogenesis.

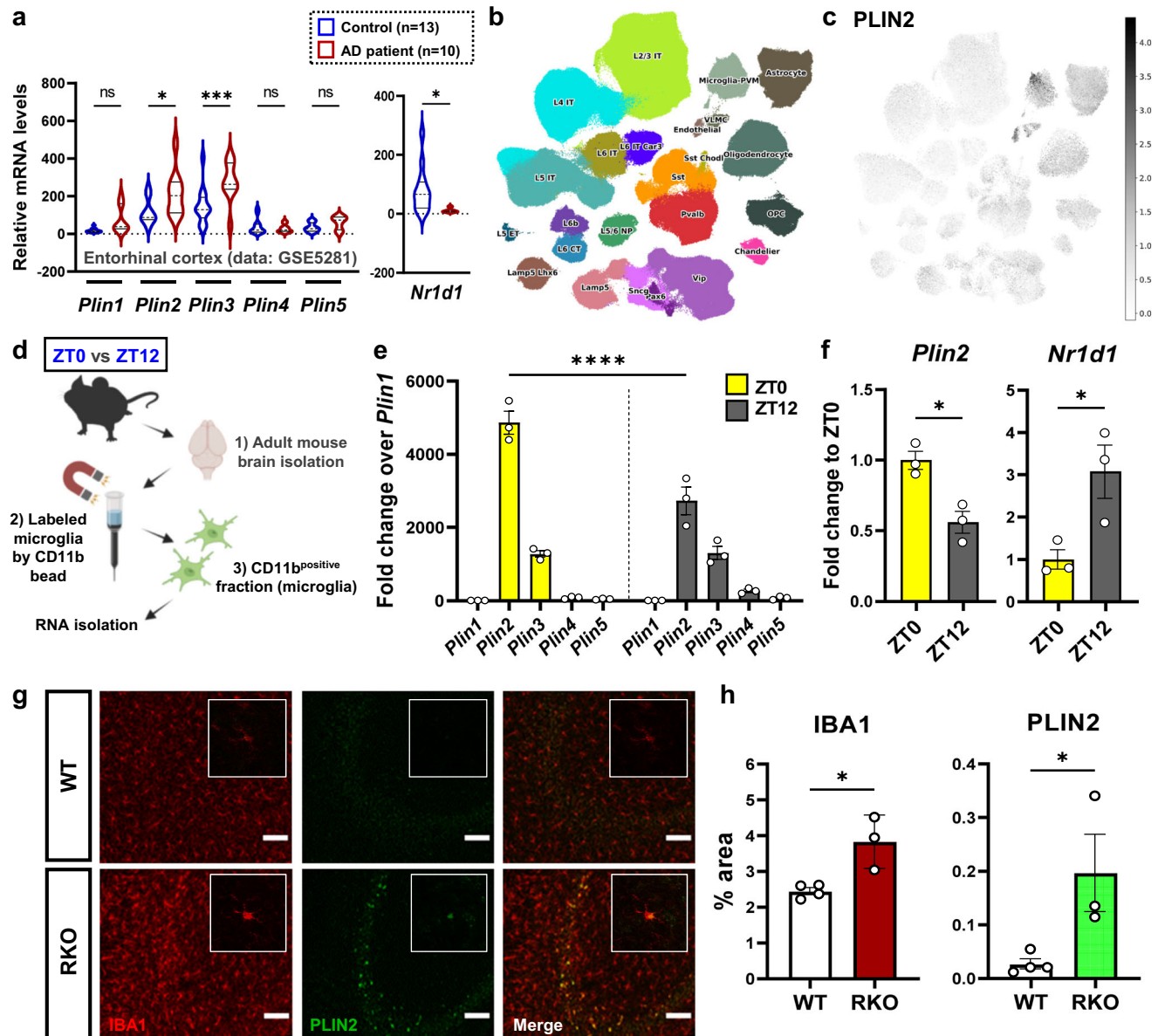

**Fig. 5 | Lipid-droplet marker Plin2 is increased in microglia in human AD patients and aged REV-ERBα KO mice. a** Transcriptional upregulation of human *PLIN2* and *PLIN3*, but not other PLIN family members, in the entorhinal cortex (EC) of AD patients, is associated with downregulation of REV-ERBα (NR1D1) in the GSE5281 dataset (control, $n = 13$; AD patient, $n = 10$). **b** UMAP of cell clusters from the Seattle AD Cell Atlas with **c** PLIN2 expression in the microglial cluster. **d** Schematic depiction of experimental workflow. Microglia were isolated from whole brain homogenate from WT mice at the two different time points (ZT0/ZT12) using CD11b. Created with Biorender.com. **e** Bar graph of each PLIN isoform and

**f** expression pattern of *Plin2* and *Nr1d1* (REV-ERBα) in isolated microglia from adult mice at ZT0 (6 am) and ZT12 (6 pm), based on mean counts-per million (CPM) from RNA-seq and were normalized to *Plin1* expression ($n = 3$). **g** Representative images and **h**, quantification of immunostaining results for microglial PLIN2 (green) and IBA1 (red) in hippocampal CA3 region from WT and global REV-ERBα KO (RKO) mice ($n = 3$–4 mice per group). Scale bar, 500 μm. *$p < 0.05$, **$p < 0.01$, ***$p < 0.005$, ****$p < 0.001$ by two-way ANOVA with Sidak multiple comparison test, two-tailed $t$ test, or one-way ANOVA. $P$ values > 0.05 are listed. Error bars represent SEM.

---

Subsequently, we hypothesized that REV-ERBα levels might influence microglial expression of PLIN2. To address this, we performed microglial isolation using adult mouse brain at lights-on (ZT0) and lights-off (ZT12) (Fig. 5d) and observed that *Plin2* and *Plin3* were the only isoforms expressed in microglia among the five PLIN isotypes (Fig. 5e). Particularly, *Plin2* had an anti-phasic expression pattern at the two-time points as compared to REV-ERBα (*Nr1d1*) (Fig. 5f), suggesting that LDs could be reciprocally regulated by REV-ERBα. To further establish this relationship, we examined microglial PLIN2 protein expression using brain sections from aged WT and germ-line global REV-ERBα KO (RKO) mice. We observed that PLIN2 protein was significantly increased in IBA1-positive microglia of RKO mice brains

including the hippocampus, cortex, and thalamus (Fig. 5g, h), suggesting that loss of REV-ERBα induces PLIN2 expression in microglia in vivo.

**Lipid-droplet accumulation impairs microglial tau phagocytosis**

AD is associated with dysfunction of lipid homeostasis in the brain, including LD accumulation in glia, and lipid metabolism is thought to be a critical driver of AD pathology[5]. Although emerging evidence supports the importance of regulating LDs in AD pathology, the exact effect of LDs on microglial responses to tau is unknown. To address this, we used oleic acid (OA), an abundant monounsaturated fatty acid, to induce the accumulation of LDs in cultured microglia (LDAM) and

measured microglia-mediated-internalization of exogenous tau aggregates. We observed that OA treatment (1 μM) induced BODIPY+ LDs (Fig. 6a) and *Plin2* transcript as well as the inflammatory markers such as *Il6* and *Il1b* in cultured microglia dose-dependently (Fig. 6b). Moreover, pharmacological inhibitors of LD formation, A922500 (20 μM) and PF-04620110 (5 μM), which both block diacylglycerol acyltransferase 1 (DGAT1) from catalyzing the final committed step in the biosynthesis of triglycerides, efficiently reduced OA-induced LD formation in microglia (Fig. 6c).

We next modified[33] and validated an in vitro tau uptake assay for measuring microglial tau internalization. FITC-labeled tau aggregates were generated by incubating human monomeric FITC-tau with 8 μM heparin for 7 days at 37 °C with shaking (Fig. 6d). We further checked aggregate morphology by electron microscopy (EM) and observed a ~200 nm filamentous form of tau after fibrilization, while monomeric tau samples were shorter than 100 nm with round shapes (Fig. 6e). To validate the effective concentration of FITC-tau aggregates for uptake assay, we performed dose-response tests using fluorescence-activated cell sorting (FACS) and determined 40 nM concentration as an optimal condition (Fig. 6f). Using this system, we measured the tau phagocytosis in OA-treated microglia with or without DGAT1 inhibitors (iDGAT1, which block LD formation) to verify the effect of LDs on microglia-mediated tau internalization. Although iDGAT1 alone had almost no effect on tau uptake, OA severely reduced internalization of FITC-tau aggregate, an effect that was partially rescued by treatment of iDGAT1 (Fig. 6g). Interestingly, treatment of microglia with LPS, which induces inflammation (Fig. 4d) and LD formation (Fig. 4e), also suppressed microglial tau uptake (Fig. S3b) and was partially recovered by iDGAT1 (Fig. S3c, S3d) with downregulation of LDs in a similar manner as OA. These results suggest that inflammation and lipid accumulation in microglia can inhibit tau uptake, and this effect is partially dependent on LD formation.

We next examined tau phagocytosis in control and REV-ERBα KO microglia cultures. Notably, REV-ERBα KO microglia had markedly decreased FITC-tau uptake under basal conditions (Fig. 6h). We observed that iDGAT1 partially reduced BODIPY+ LDs (Fig. 6i), and caused a small but statistically-significant recovery in tau phagocytosis of REV-ERBα KO microglia (Cre+) (Fig. 6j). These data suggest that microglial REV-ERBα deletion may exacerbate tauopathy in part via impairing microglial clearance of tau aggregate, and that lipid accumulation and LDs may contribute to this.

## Altered sphingolipid metabolism is a conserved feature of microglia following tau exposure or REV-ERBα deletion

To further understand the relationship between tau pathology, microglial lipid homeostasis, and microglial tau clearance, we developed a protocol for the preparation of Tau-enriched brain extract (TBE) from aged (18 month) P301S PS19 tau mutant mouse brain (Fig. S4a). We confirmed the presence of both total human tau (hTau) and phospho-tau (pTau) in TBE, but not in similar extracts from WT brains (Fig. S4b) and quantified the concentration of pTau in TBE by ELISA (Fig. S4c). Next, we performed mass spectrometric proteomic analysis to verify how much tau protein is contained in TBE, and what other proteins are present. We observed that 'Microtubule-associated protein tau' was 51st hit with 57% of sequence coverage among total 2922 identified proteins (Fig. S4d–S4f).

To determine whether TBE triggers LD formation in microglia, we performed BODIPY dye staining after treatment with TBE in cultured microglia, and found that TBE efficiently increased BODIPY+ LD expression (Fig. S5a) and *Plin2* transcript dose-dependently (Fig. S5b), along with large increases in pro-inflammatory cytokine transcripts including *Il6* and *Il1b* (Fig. S5b). Moreover, iDGAT1 prevented LD accumulation (Fig. S5c) and PLIN2 expression (Fig. S5d, S5e) evoked by TBE in microglia. Finally, TBE treatment reduced microglial-mediated

tau internalization in a dose-dependent manner (Fig. S5f), showing that TBE induces a phenotype similar to that seen with microglial REV-ERBα KO.

To explore which lipid species were altered by TBE treatment in microglia, we performed lipidomic analysis using vehicle (VEH) or TBE-treated microglial lysate. Interestingly, 'sphingolipids' was identified as the most increased lipid class (45%) from among 457 classified lipids in microglia (Fig. 7a). 35% 'glycerophospholipids', 15% 'glycerolipids', and 5% 'organonitrogen compounds' were also included in TOP 20 analysis (Fig. 7a) and individual information of each class were presented in Fig. 7b. Specifically, palmitoyl-sphingomyelin (pSM) features were dominantly induced by 5–8-fold in TBE-treated microglia as compared to control cells (Fig. 7c). The observed MS1/MS2 spectrum supported pSM signature in TBE-treated microglia (Fig. 7d, e). We observed that acute treatment of microglia with pSM also led to abnormal lipid accumulation (Fig. 7f). Since higher levels of 'sphingomyelins (SMs)' species such as SM d18:1/22:1 and d18:1/26:1 have been described in the prefrontal and entorhinal cortices of AD patients[34], our observation supports and highlights the possible importance of modulating abnormal levels of SM in microglia to preserve microglial phagocytic activity.

To better understand whether metabolic changes in REV-ERBα KO microglia could be related to abnormal sphingomyelin metabolism, we surveyed the expression levels of sphingomyelin-related enzymes from our RNA-seq data from Fig. 4. REV-ERBα KO microglia (Cre+) had significant induction of sphingomyelin synthase 2 (*Sgms2*) and sphingomyelin phosphodiesterase 2 (*Smpd2*), suggesting that metabolism of sphingomyelin in microglia was altered by REV-ERBα KO, and that REV-ERBα KO microglia may have similar lipid metabolic changes as TBE-treated microglia (Fig. 7g).

## REV-ERBα deletion amplifies TBE-induced microglial LD accumulation in a sex-dependent manner

As we observed sex-dependent effects of microglial REV-ERBα deletion on tauopathy in PS19 mice (Figs. 2, 3), we hypothesized that the effects of microglial REV-ERBα KO on LD accumulation and microglial tau phagocytosis might vary in a sex-dependent manner in the presence of tau. To test this possibility, we treated male and female REV-ERBα KO microglia (Cre+) and Cre− control cells with TBE and measured LD levels by microscopy and flow cytometry. Unlike the non-TBE-treated condition (Fig. 4), REV-ERBα KO amplified TBE-dependent LD accumulation only in male microglia, but not in female microglia (Fig. S6a–S6c). We next examined tau uptake in REV-ERBα KO (Cre+) and Cre− control male microglia treated TBE with or without LD blocker, iDGAT1. In both Cre− and Cre+ cells, TBE strongly suppressed tau uptake to a similar degree, and tau uptake was not rescued by iDGAT1 treatment (Fig. S6d). These finding suggest that microglial REV-ERBα modulates TBE-induced LD formation in a sex-dependent manner, but that the effect of TBE on tau uptake is not impacted by REV-ERBα or DGAT1. We suspect that microglial REV-ERBα could help enhance tau phagocytosis early in disease progression based on the results in Fig. 4, but this pathway is likely unable to restore tau phagocytosis in the setting of severe tau pathology later in disease. Overall, we conclude that REV-ERBα KO inhibits tau uptake in microglia via a mechanism that relies in part on LD accumulation and can be partially recovered by LD blockers. However, TBE causes microglial dysfunction and impaired tau internalization through other pathways which cannot be overcome with LD blockade.

In summary, our current work shows that REV-ERBα deletion causes altered lipid metabolism and inflammatory signaling in microglia, leading to abnormal accumulation of LDs, impaired microglial phagocytic clearance of tau, increased microglial inflammatory responses, and ultimately exacerbated tau pathology in vivo (Fig. 8). These effects are sex-dependent and most prominent in male mice.

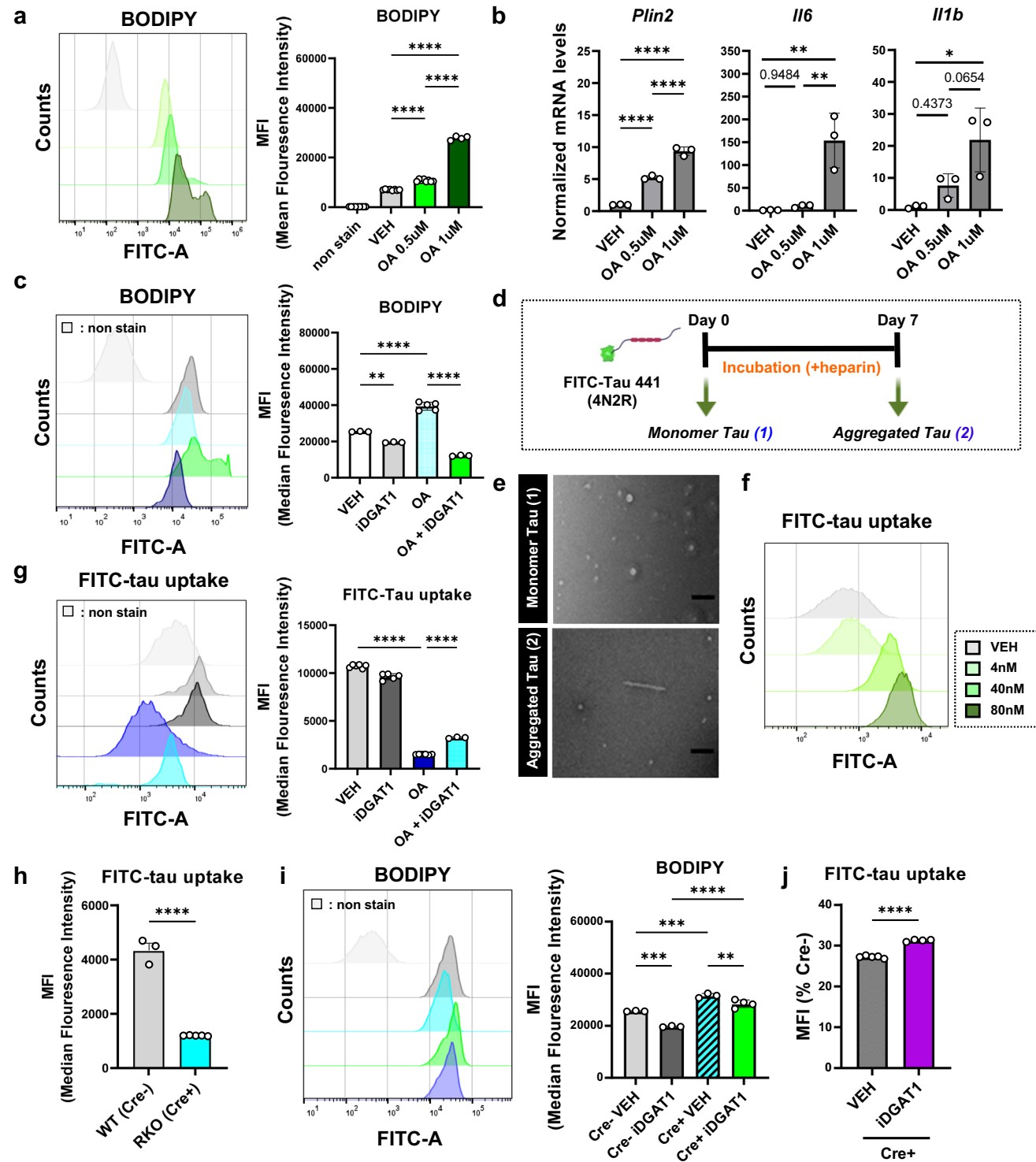

**Fig. 6 | Lipid accumulation contributes to impaired tau uptake in REV-ERBα KO microglia. a** Dose-dependent increase in BODIPY+ LDs after oleic acid (OA, 1 μM) treatment in cultured microglia ($n = 4$–9 independent cell cultures). **b** Dose-dependent increase in pro-inflammatory cytokine (*Il6* and *Il1b*) and *Plin2* gene expression in cultured microglia after OA treatment ($n = 3$ independent cell cultures). All transcript levels are normalized to DMSO-treated cells (VEH). **c** Pharmacological inhibitors of LD formation (iDGAT1; 10 μM A922500 and 5 μM PF-04620110) block increases in BODIPY signal in OA-treated cultured microglia by flow cytometry ($n = 3$–6 independent cell cultures). **d** Strategy for generating FITC-tau aggregates. Monomeric form, sample (1), is incubated for 7 days with heparin (10 mM) on 37 °C shaking incubator for sample (2) and centrifugation, to isolate pure aggregate. Created with Biorender.com. **e** Electron micrograph comparing the

structure of monomeric and fibril forms of FITC-Tau. Scale bar: 100 nm. **f** Validation of optimal concentration of FITC-tau aggregates for uptake assays based on dose-response curve (4 nM, 40 nM, and 80 nM) in cultured microglia using flow cytometry. **g** Inhibition of FITC-tau uptake by OA is partially rescued by inhibitors of LDs formation ($n = 3$–6 independent cell cultures). **h** REV-ERBα KO (Cre+) microglia show decreased FITC-tau uptake compared to Cre− controls after 2 hours incubation. **i** iDGAT1 inhibitors only partially prevent LDs accumulation (BODIPY+ signal) in Cre− control and in REV-ERBα KO microglia. **j** iDGAT1 inhibitors partially rescue FITC-tau uptake in Cre+ REV-ERBα KO microglia. Data presented as % of untreated Cre− cell FITC-tau uptake. *$p < 0.05$, **$p < 0.01$, ***$p < 0.005$, ****$p < 0.001$ by one-way ANOVA or two-way ANOVA with Sidak test or two-tailed $T$ test. $P$ values > 0.05 are listed. Error bars represent SEM.

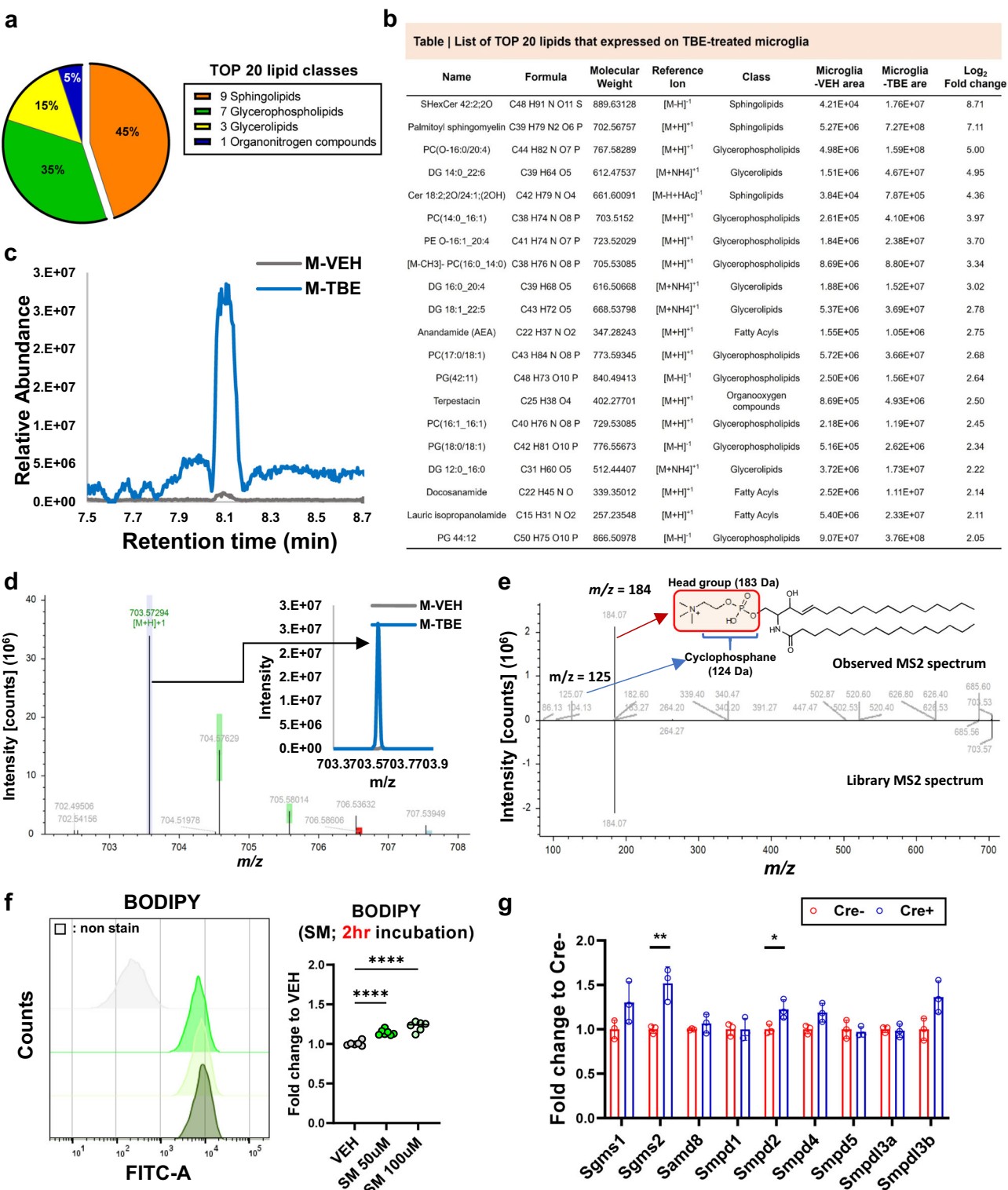

**Fig. 7 | Sphingomyelin is ranked as the dominant lipid feature in TBE-treated microglia and sphingomyelin metabolism is altered in REV-ERBα KO. a** Pie chart showing that TOP 20 lipid classes which are the most upregulated in TBE-treated microglia compared to VEH-treated controls. **b** Top 20 differentially expressed lipids in TBE-treated microglia based on Log₂ fold change. **c** The extracted chromatogram of phosphatidyl sphingomyelin (pSM) (m/z = 703.5729, [M+H]+) compared between VEH− and TBE-treated microglia **d** Observed MS1 spectrum of pSM. **e** Observed MS2 spectrum of pSM. Signature fragmentation of the head group (183 + H) at m/z = 184 and cyclophosphane (124 + H) at m/z = 125 was indicated with red and blue arrow, respectively. **f** BODIPY+ signals are accumulated in microglia by pSM treatment in a dose-dependent manner ($n = 6-7$ independent cell cultures). **g** Upregulating sphingomyelin metabolic process in REV-ERBα KO microglia ($n = 3$). *$p < 0.05$, **$p < 0.01$, ****$p < 0.001$ by one-way ANOVA or two-way ANOVA with Sidak test. $P$ values > 0.05 are listed. Error bars represent SEM.

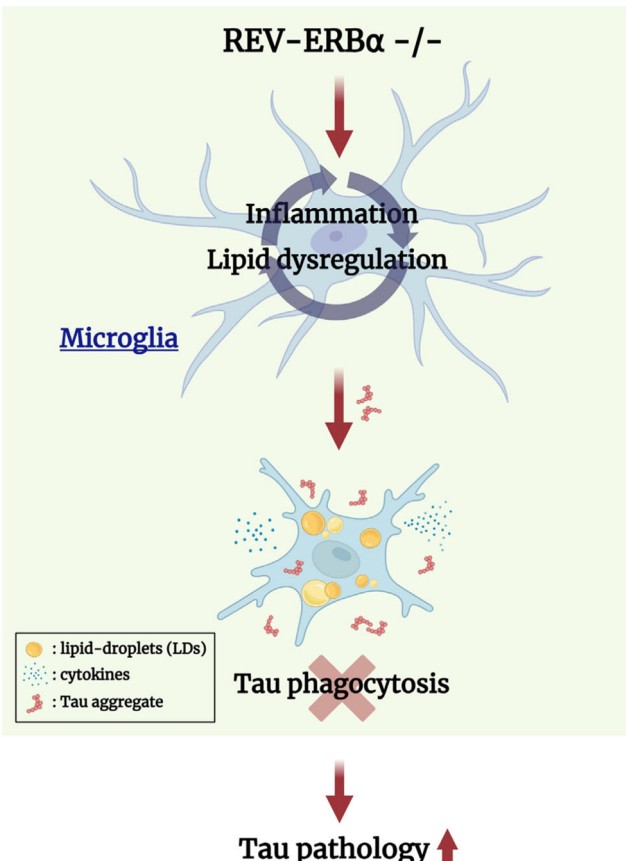

**Fig. 8 | Hypothetical mechanism of REV-ERBα on microglia-mediated internalization of tau aggregatse through control of lipid-droplet expression and its effect on tauopathy in male PS19 mice.** Microglial REV-ERBα KO leads to LD accumulation and inflammation in microglia and causes dampened microglia tau phagocytosis. Tauopathy is more severe in microglial REV-ERBα KO; PS19 mice than in control PS19 mice, particularly in male mice. Created with Biorender.com.

## Discussion

Tau aggregation in the form of neurofibrillary tangles is a well-known pathological hallmark of AD and is emerging as a popular therapeutic target[21]. Although tau is predominantly expressed in neurons, aggregated forms can be removed by glia such as astrocytes and microglia to prevent its accumulation in the brain. Recently, lipid-droplet accumulating microglia (LDAM) have emerged as a new microglia subpopulation in neurodegenerative diseases including AD[5] and Parkinson's disease (PD)[35,36]. However, it is not well known how LDAM affect tauopathy. Here, we have presented data showing that the circadian nuclear receptor REV-ERBα can regulate lipid metabolism in microglia and tau deposition in the brain in mice. We provide evidence that microglial REV-ERBα regulates inflammatory tone and lipid metabolism to affect microglial phagocytic clearance of tau and suggest that this regulatory mechanism may affect tau pathology specifically in male PS19 mice. Moreover, we observed tau-enriched brain extract induces both inflammatory signaling and lipid metabolic changes in microglia, characterized by sphingolipid accumulation, that can impair tau phagocytosis. Interestingly, REV-ERBα KO microglia may also have dysfunction of sphingomyelin metabolism, indicating a possible common mechanism linking sphingomyelin accumulation to impaired tau phagocytosis. These results suggest that modulating microglial lipid homeostasis and lipid droplet formation, perhaps through manipulation of REV-ERBα, could be a therapeutic strategy for AD and other tauopathies.

Several studies have reported abnormal lipid accumulation and enrichment of LDs in microglia during aging and in neurodegenerative diseases including AD[5,6,37]. However, how abnormal LDs in microglia influence AD pathology remains unclear. We observed that REV-ERBα KO shifts microglia toward an LDAM-like phenotype with extensive LD accumulation (Fig. 4f–h), altered lipid metabolic gene expression (Fig. 4b), impaired tau uptake (Fig. 6i), and ultimately more severe tau pathology in vivo, particularly in male mice (Figs. 2, 3). Since LPS-stimulated inflammatory microglia also show LD accumulation and defective internalization of tau with LD accumulation (Fig. S3 and Fig. 4e), we suspect that microglial REV-ERBα KO exacerbates tau both by increasing inflammation, and by altering lipid homeostasis, and that these processes interact. Our RNA-seq data show cell activation and lipid metabolism as top dysregulated processes in REV-ERBα KO microglia, supporting this hypothesis. In WT microglia, lipid exposure (OA) alone promoted LDs, increased the expression of inflammatory cytokines, and abrogated microglial tau phagocytic function (Fig. 6), showing that the relationship between LDs, inflammation, and tau uptake in microglia is tight and that these processes may have synergic effects. Microglial tau uptake is deficient in REV-ERBα KO microglia and only partially rescued by blockage of LD formation, suggesting that other pathways, such as inflammation, may also contribute. Thus, to improve the defects of microglial tau uptake in REV-ERBα KO microglia on tau uptake, LD-mediated inflammatory responses should be highlighted in future studies.

Our RNA-seq data show that REV-ERBα KO microglia have multiple dysregulated lipid metabolic pathways. However, the exact molecular mechanism by which REV-ERBα KO causes LD accumulation is still unknown. We observed that isolated microglia specifically express *Plin2* in a time-of-day dependent manner which is reciprocal with REV-ERBα (Fig. 5e, f), and ChIP-seq data from the brain of HFD (high-fat diet fed) mice shows direct binding of REV-ERBα on an RRE site in the *Plin2* promoter region, a known consensus sequence for REV-ERBα binding for transcriptional repression[38]. In the absence of REV-ERBα, microglial *Plin2* transcript is significantly increased (Fig. 4i). These findings suggest that REV-ERBα could directly regulate *Plin2* expression to influence LD formation. However, our data show that under basal conditions in culture, *Plin2* knockdown does not prevent LD formation in microglia (Fig. S7). Thus, it is unclear if changes in *Plin2* in REV-ERBα cells are a major cause or effect of changes in LD content, particularly in vivo and under pathological conditions. Moreover, the role of *Plin2* in LD formation is also not clear. Further investigation is needed to determine which genes drive LD formation in REV-ERBα KO microglia, as these could be targeted to overcome abnormal LD expression under pathological conditions.

Although sphingolipids are thought to be biologically relevant biomarkers of AD[39], they have not yet been investigated in microglia, one of the major cell types that accumulates LDs in disease. In our studies, we provide lipidomic analysis showing that sphingomyelin (SM), particularly palmitoyl-sphingomyelin (pSM), prominently accumulates in tau-brain extract (TBE)-treated microglia and suggest it as a possible therapeutic target for AD according to its abundance after tau treatment. Since TBE also induces LDs, inflammation, and impairs tau phagocytic activity in microglia (Fig. S5), SM metabolism should be investigated further in future studies as a possible mediator of microglial dysfunction and tau pathogenesis in AD. Notably, REV-ERBα KO microglia also show transcriptional changes consistent with abnormal sphingomyelin metabolism, as well as these other features, indicating that SM may be a common link, and that the effects and mechanisms of SM in microglia in AD should be studied further.

The therapeutic implications of our findings are very different depending on the phase of AD. According to our observations, blocking LD accumulation partially recovers microglial tau uptake activity in the setting of OA treatment or REV-ERBα KO, but not in TBE-treated conditions (Fig. S6d). This suggests that lipid modulation

therapy might not be effective later in the disease course when there is extensive tau pathology present, and should be considered at the early stage of AD. In addition to lipid metabolism, energy metabolism including glycolysis, glucose, and oxidative stress also sensitively respond to tau oligomer treatment in microglia[40], suggesting that combination therapy targeting multiple aspects of microglial metabolic function may be needed for late phase of AD. Likewise, our RNA-seq data shows that REV-ERBα KO in microglia impacts diverse metabolic changes such as chemotaxis, cell activation, RNA processing, and receptor-mediated endocytosis. These suggest that additional studies are needed to understand the mechanisms by which REV-ERBα affects microglial tau uptake. For example, heparan sulfate proteoglycans (HSPGs) are involved in tau phagocytosis[41], and a previous report shows that circadian HSPG expression mediates rhythmic phagocytosis of amyloid-β (Aβ)[42], though this has not been shown for tau. Neuronal uptake of tau aggregates requires HSPG-mediated micropinocytosis, so perhaps this mechanism applies in microglia as well and could be under control of REV-ERBα[43]. TREM2-dependent LD biogenesis through PLCγ2[44] could be another candidate mechanism by which REV-ERBα could influence tau uptake in microglia.

Our findings show that REV-ERBα KO exacerbates TBE-induced LD accumulation only in male microglia and exacerbates tau pathology in two in vivo models only in males, suggesting that the sex-specific effects of REV-ERBα on microglial metabolism go beyond inflammatory signaling. Recent transcriptomic and proteomic studies have shown that male and female microglia have differential gene expression profiles at the baseline and respond differently to inflammatory stimuli[16]. Female microglia have anti-inflammatory characteristics and can mitigate local injury when transplanted into male mice, but male microglia cannot[16]. Proteomic analyses show that male microglia express significantly more Toll-like receptor (TLR) pathway-related proteins than females, whereas Interferon regulatory factor 3 (Irf3) is highly enriched in female microglia, indicating that both microglia might respond differently to the same stimuli[15]. These observations provide insight into why REV-ERBα KO in microglia amplifies this sex-dependent inflammatory dichotomy between male and female microglia, exacerbating tau- (TBE) and LPS-mediated inflammatory responses, tau-mediated LD formation, and tau pathology in male mice only, both in vitro and in vivo. In addition, previous studies have shown circadian regulation of IRF target genes and proteins, providing another possible mechanism by which REV-ERBα could exert sex-specific effects in microglia[45,46]. However, further work is needed to understand sex-specific pathways in microglia in general, and their regulation by REV-ERBα and other clock proteins.

Our previous work demonstrated that germ-line global REV-ERBα KO mice developed spontaneous microglial and astrocyte activation and inflammation[4]. However, in this study, we did not observe severe spontaneous neuroinflammation in our microglia-specific, post-natal REV-ERBα KO mice, and only found slight astrocyte activation (Fig. 1). Thus, the more severe inflammatory phenotype observed in germ-line global KO mice must be the result of deletion of REV-ERBα in multiple cell types/tissues simultaneously, or a developmental effect. Certainly, REV-ERBα exerts effects in peripheral organs that could impact brain inflammation, and global deletion could also potentially cause neuronal injury which primes glial inflammatory responses. Moreover, global germ-line deletion of REV-ERBα KO also mitigated amyloid pathology in 5xFAD mice and enhanced microglial phagocytosis of Aβ[4]. It is unclear if these disparate results between Aβ and tau models suggest that REV-ERBα KO regulates pathology-specific responses, or if these differences are due to the use of global germ-line deletion versus inducible microglia-specific knockout. Specific molecules mediating microglial uptake of Aβ and tau are different[43], so it remains possible that REV-ERBα regulates Aβ-specific phagocytic machinery in one direction and tau-specific pathways in the other. Further studies in inducible global KO mice, as well as other cell and tissue-specific KO

lines, will be needed to exclude a developmental effect, or to dissect a complex multicellular response. This is an important issue, as REV-ERBα-targeted pharmaceuticals are being developed and could have complex effects. Moreover, since REV-ERBα is a highly circadian protein in most tissues including brain, elucidating its full effects across the cell types is critical for understanding how circadian disruption may mediate disease through the alteration of REV-ERBα rhythms.

Our study has several additional limitations. We were not able to examine behavioral changes related to the pathological changes seen in our mice. The effects of microglial REV-ERBα deletion on neuronal function, in terms of synaptic plasticity and electrophysiology, were also not examined, and remain unknown. However, the increased tau aggregation and neuronal loss in CA1 that we have shown are indicative of neuronal damage and are associated closely with neuronal dysfunction and cognitive impairment in mice and humans[19]. The specific molecular underpinnings of the sex differences observed are also not yet understood, though the same can be said for sex differences in AD in general.

In summary, our findings provide insights into the regulation of LD formation in microglia, implicating REV-ERBα as a regulator of this process. We demonstrate the impact of microglial REV-ERBα and LDs formation in general on tau metabolism and pathology, revealing an interaction between REV-ERBα, lipid homeostasis, microglial function, and tauopathy. These findings illustrate the importance of modulating REV-ERBα activity as a potential therapeutic target for AD, as well as a means of reducing disease-related microglia dysfunction across neurodegenerative conditions.

## Methods

### Mice
All mouse experiments were performed according to protocols approved by the Washington University IACUC (Institutional Animal Care and Use Committee, Office of Laboratory Animal Welfare Assurance: D1600245, USDA Registration #43-R-008) and under the supervision of the Department of Comparative Medicine. REV-ERBα global knockout (RKO), 5XFAD, CAG::Cre^ERT2+, CX3CR1::Cre^ERT2+, and MAPT P301S mice with C57/Bl6J background were obtained from The Jackson Laboratory. REV-ERBα floxed mice were kindly provided by Dr. Lazar (University of Pennsylvania, Philadelphia, USA)[47]. All mice were housed under 12:12 hr light-dark cycle and free access to water and food. CX3CR1::Cre^ERT2+; Nr1d1^fl/fl mice were given tamoxifen (Sigma, 2 mg/day for 5 days) by oral gavage at 2 months of age to induce REV-ERBα deletion. Unneeded mice were sacrificed by $CO_2$ euthanasia according to the approved protocol of Washington University IACUC.

### Immunohistochemistry
Mice were deeply anesthetized by i.p injection with Fatal Plus pentobarbital and subjected to thoracotomy and transcardiac perfusion with cold dPBS containing 0.3% heparin. Brain tissue was then either dissected and flash-frozen or drop-fixed in 4% paraformaldehyde solution for 24 hours. Fixed hemispheres were then incubated with 30% sucrose for at least 36 hours at 4 °C and then 40 μm thickness frozen sections were prepared on a Leica sliding microtome. All sections were stored in cryoprotectant solution at −20 °C prior to use.

For X34 plaque staining, sections were permeabilized with 0.25% Triton X-100 (PBSX) for 30 min at room temperature (RT) and then incubated with X34 solution (Sigma-Aldrich) which was diluted 1:1000 in staining buffer at RT. After 20 min, sections were washed twice with X34 wash buffer for 2 min, and then washed three times with PBS for 5 min.

For fluorescence staining, sections were blocked and permeabilized with 3% donkey serum in 0.4% Triton X-100 (PBSX) and stained with the following antibodies: mouse anti-NeuN (Sigma-Aldrich, MAB377, 1:200), mouse anti-GFAP (Novus, 5C10, 1:2000), rabbit anti-IBA1 (Wako, 019-19741, 1:1000), rat anti-CD68 (Bio-rad, MCA1957,

1:500), guinea pig anti-Plin2 (Fitzgerald, 20R-AP002, 1:200). Sections were incubated with primary antibodies in 0.4% Triton X-100 (PBSX) containing 1% donkey serum at 4 °C overnight. Sections were then washed three times with PBS and incubated with secondary antibodies which were diluted 1 to 400 in 0.25% Triton X-100 (PBSX) for 1 h at RT. Sections were washed three times with PBS and mounted on slides with Prolong Gold.

## AT8 staining

Brain sections were washed three times with Tris-buffered saline (TBS) for 5 min and incubated with 0.3% hydrogen peroxide ($H_2O_2$) in TBS for 10 min at RT. Sections were then washed three times with TBS for 5 min and blocked with 3% skim milk in 0.25% Triton X-100 (TBSX) for 30 min, and then incubated with biotinylated AT8 antibody (Thermo Fisher Scientific, MN1020, 1:1000) in 0.25% TBSX containing 1% skim milk at 4 °C overnight. After three times washes with TBS for 5 min, the sections were incubated with ABC Elite (Vector PK-6100, 1:400) in TBS for 1 h and were visualized using 3,30−diaminobenzidine tetra-hydrochloride (DAB) (Chromogen).

## AT8 ELISA

The concentrations of pTau in tau-enriched brain extract (TBE) were measured using sandwich ELISA with the AT8 antibody (Thermo Fisher Scientific). 96-well costar ELISA clear plate (Thermo Fisher Scientific, #3690) was coated with 20 µg/ml of AT8 antibody in carbonate coating buffer (PH 9.6). After overnight incubation at 4 °C, the plate was five times washed with PBS and blocked with 190 µl of 4% BSA in PBS for an hour at RT with shaking. Five times washes were followed by then, 50 µl of serial diluted standards of human AD tau, and sequential diluted TBE samples were incubated overnight at 4 °C. After eight times washes with PBS, 50 µl of biotinylated AT8 (Thermo Fisher Scientific, 1;500) with 1% BSA in PBS was incubated at RT for 1.5 hrs and washed eight times with PBS. Strep-poly-HRP40 (Fitzgerald, 1:6000) with 1% BSA in PBS was incubated for 1.5 hrs in the dark with shaking and incubated 50 µl of super slow TMB substrate (Thermo Fisher Scientific) for 5–15 min after eight times washing with PBS. Kinetics were measured at 650 nm using plate reader (Bio-Tek Synergy 2).

## Isolation of microglia from adult mouse brain

Mice were anesthetized by Fatal Plus pentobarbital with i.p injection and perfused with 30 ml of ice-cold dPBS. Brains were minced well using razor blade in a petri dish filled with 10 ml digestion buffer (0.05% collagenase D, 1 µg/ml TLCK trypsin inhibitor, 20 U/ml DNase 1, 10 mM pH7.4 Hepes in HBSS) and were incubated in a conical tube at 37 °C water bath for 25 min with inverting every 10 min. Dissociated brain tissue was filtered through 70 µm cell strainer and rinsed two times with ice-cold dPBS. Cells were pelleted in ice-cold dPBS after centrifugation at $300 \times g$ for 5 min, then resuspended and mixed with debris removal solution. The top layer was gently covered with 4 ml of ice-cold dPBS. Sample was centrifuged at $3000 \times g$ for 10 min with full acceleration and brake on 2 at 4 °C. Among the three resulting phases, the two top layers were completely discarded, and the tube filled with 15 ml of ice-cold dPBS. Sample was centrifuged with full acceleration and brake at $1000 \times g$ for 10 min after three inversions. Cell pellet was resuspended with 1 ml of cold 1× Red Blood Cell Removal Solution and was incubated for 10 min at 4 °C. After adding 10 ml of ice-cold MACS buffer (0.5% BSA, 8 mM EDTA in dPBS), sample was again centrifuged at $300 \times g$ for 7 min and the supernatant discarded. The cell pellet was resuspended with 1 ml MACS buffer and filtered with 70 µm cell strainer and incubated with 10 µl of CD11b microbeads (Miltenyl Biotec, 130-049-601) for 15 min at 4 °C. Cells were then washed with 1–2 ml of MACS buffer and centrifuged at $500 \times g$ for 5 min. CD11b-magnetically labeled cells were then isolated using a MACS separator according to manufacturer's instructions.

## Cultured primary microglia

Microglia were separated from mixed glial cultures that were obtained from postnatal days 1–3 (P1-3) mouse pups. Dissected cortical regions were isolated and the meninges removed. Tissue was then trypsinized with 0.05% trypsin-EDTA for 15 min at 37 °C. Cells were neutralized with two volumes of complete media (DMEM + 10% FBS) centrifuged for 5 min. After discarding of the supernatant, cells were resuspended with full volumes of complete media, plated on Poly-ᴅ-Lysine (PDL)-coated T75 flasks, and grown to DMEM + 10% FBS. Media was changed to include GM-CSF (5 ng/ml) containing media two days later and kept for 10 days. Floating microglia were collected from the mixed glia by shaking the flask at 225 rpm for 2 hrs.

## Sex determination

Sex was confirmed by tail lysate of each pup and male-specific marker SRY (sex determination region on the Y chromosome responsible for testes formation) and the universal marker myogenin (Myog) which expressed in both males and females were amplified by polymerase chain reaction (PCR). The PCR Primer sequences were; Sry, 5′-TCAT-GAGACTGCCAACCACAG-3′ and 5′-CATGACCACCACCACCACCAA-3′. Myog,5′-TTACGTCCATCGTGGACAGC-3′ and 5′-TGGGCTGGGTGTTA GTCTTA-3′. PCR condition was: 95 °C for 3 min and then 30 cycles of denaturation at 95 °C for 15 s, annealing at 58 °C for 15 s, and elongation at 72 °C for 1 min. Following 30 cycles, the reaction concludes with a final 72 °C elongation for 1 min. Each product size was 441 bp (Sry) and 245 bp (Myog), respectively.

## Cell culture

Cultured primary microglia were grown in DMEM with 10% fetal bovine serum (FBS) (Gibco) and 100 U/ml Penicillin-Streptomycin (Gibco). All cells were maintained at 37 °C in a humidified atmosphere with 5% CO2. For *lipopolysaccharide* (LPS) studies, we treated 100 ng/ml of LPS (Sigma, L6529) on cultured primary microglia for 24 hrs. 4-hydroxytamoxifen was added to culture media at 1.5 µM for 24 hours to induce deletion of endogenous REV-ERBα in cultured microglia from CAG::Cre$^{ERT2}$;*Nr1d1*$^{fl/fl}$ mice.

## Immunocytochemistry and BODIPY staining

Cultured cells were seeded on PDL-coated glass coverslips and fixed with 4% paraformaldehyde (PFA) for 15 min. Each well was then washed two times with PBS and permeabilized/blocked with 10% horse serum (HS) in 0.3% Triton X-100 (PBSX) for 30 min at RT. After two washes with PBS, the samples were incubated with the following primary antibodies diluted in 0.1% Triton X-100 (PBSX) overnight at 4 °C; rabbit anti-IBA1 (Waco, 019-19741, 1:500) and guinea pig anti-PLIN2 (Fitzgerald, 20R-AP002, 1:200). The cells were then washed three times with PBS and incubated with secondary antibodies which were diluted 1–500 in 0.1% Triton X-100 (PBSX) for 30 min at RT. The nuclei were stained with 4′,6-diamidino-2-phenylindole (DAPI) for 10 min and were washed with PBS. For BODIPY staining, BODIPY™ 493/503 (Invitrogen, D3922, 1:1000) dye was mixed with DAPI staining solution and incubated with cells for 10 min. Coverslips were mounted with fluorescence mounting medium (ProLong Gold antifade reagent, Invitrogen, P36934).

## RNA preparation and qPCR analysis

Total RNA was extracted from cells using PureLink RNA mini kit (Invitrogen) and the concentration of each RNA was determined by Nanodrop ND 1000 spectrophotometer. cDNA was then synthesized using approximately 1 µg of RNA using high-capacity cDNA Reverse Transcription Kit (Life Technologies) according to the manufacturer's instructions. Real-time quantitative qPCR was performed using ABI TaqMan primers and ABI PCR Master Mix buffer on ABI StepOnePLus or QuantStudio 3 thermocyclers (Applied Biosystems). The qPCR Primers were purchased by Thermo Fisher and the information were;

Cd68, Mm03047343_m1; Aif1, Mm00479862_g1; Il1b, Mm00434228_m1; Il6, Mm00446190_m1; Tnfa, Mm00443258_m1; Gfap, Mm01253033_m1; Nr1d1, Mm00520711_m1; C1qa, Mm00432142_m1; Cre, Mm00635245_cn; Plin2, Mm00475794_m1; Actn, Mm02619580_g1.

## Transmission electron Microscopy

Electron microscopy experiments were carried out by the Washington University Center for Cellular Imaging. For negative-stain specimens, 10 μl of each sample was deposited onto a freshly glow-discharged carbon-coated EM grid (Ted Pella). After 1 min incubation at room temperature, grids were washed by swirling for 2 seconds through five drops of ultrapure water, and one drop of 0.75% aqueous uranyl formate (Electron Microscopy Sciences), then placed on a fresh drop of 0.75% uranyl formate for 1 min. The grids were then blotted using a filter paper to remove excess solution and dried before imaging. Negative-stain images were acquired on a TEM (Jeol JEM-1400 Plus) at 120 kV.

## Intracerebroventricular P0 injection

Postnatal day 0 (P0) pups were anesthetized on ice for 3 min and then received intracerebroventricular (ICV) injections of 2 μl of AAV (1.3E + 10 viral particles/ventricle; 2 μl/ventricle) into the lateral ventricles using a 30-gauge needle. AAV1-Tau$^{P301L}$ was provided Dr. Leonard Petrucelli[21]. At 4 or 6 months of age, the whole brain was removed and fixed with 4% PFA overnight at 4 °C for histology.

## Preparation of FITC-tau aggregate

A total of 100 μg of Fluorescein (FITC) Tau 441 peptide (rPeptide) were pre-incubated with 10 mM 1,4-dithiothreitol (DTT), 100 mM NaCl, and 10 mM HEPES in water for 1 hr at RT and then heparin was added at a final concentration of 8 μM. After 7 days of incubation at 37 °C with shaking, samples were centrifugation at $100,000 \times g$ for 1 hr at 4 °C. The pellet was considered as aggregated tau and sonicated with 65% power/amplitude for 30 sec in a water bath. Tau aggregates were then resuspended and diluted from use.

## Tau uptake assay

Cultured microglia were plated at 10,000 cells per well in a 12-well plate. The next day the medium was replaced, and cells were treated with 40 nM FITC-tau aggregate for 2 h at 37 °C. Cells were then washed twice with PBS and trypsinized with 0.25% trypsin-EDTA. After neutralization with complete culture media (10% FBS in DMEM), the samples were centrifuged at 1250 rpm for 5 min at 4 °C and the supernatant discarded. The pellet was washed through resuspension in ice-cold PBS, followed by centrifugation again with the same conditions. The pellet was resuspended with FACS buffer (5% FBS in dPBS) and subjected to flow cytometry (BD Biosciences). About 10,000 cells per sample were analyzed and the data were presented using FlowJo software (BD Biosciences).

## Tau-brain extract (TBE) purification

Mice were anesthetized by Fatal Plus pentobarbital with i.p injection. The brain was rapidly removed and frozen with isopentane on dry ice. Frozen brain was then homogenized with buffer H (10 mM Tris-HCl pH7.4, 0.8 M NaCl, 1 mM EDTA, 2 μM DTT, cOmplete, PhosStop, 0.1% sarkosyl, 10% sucrose) with homogenizer until all large chunks are gone. The samples were homogenized again with insulin syringe more than five times and freeze-thaw once on dry ice. After then, the samples were centrifuged at $10,000 \times g$ for 10 min at 4 °C, and supernatant were removed and kept as 'S1' on ice. One volume of buffer H was added to the pellet and re-homogenized with sonicator with 30% amplitude, 1 s/1 s pulse for 1 min. Homogenized samples were centrifugation with $10,000 \times g$ for 10 min at 4 °C. Supernatant from this were considered as 'S2' and pooled with 'S1'. Sarkosyl was added into

the 'S1 + S2' resulted in 1% as a final concentration and incubated for 1 hr at RT with shaking. Incubated samples were then centrifuged with $300,000 \times g$ for 1 hr at 4 °C and took supernatant to use as soluble fraction (SS). Total tau and phosphorylated tau contents were further validated by western blotting.

## Immunoblotting

TBE sample was separated by electrophoresis on 4–12% sodium dodecyl sulfate–polyacrylamide gels and then transferred electrophoretically to polyvinylidene difluoride membranes. The membranes were blocked with 5% skim milk and then washed three times with 0.1% Triton X-100 (TBSX). The membranes were then gently agitated and incubated at 4 °C overnight with the following primary antibodies: HT7 (Thermo Fisher, MN1000, 1:500), AT8 (Thermo Fisher, MN1020, 1:1000), and Plin2 (Abcam, ab52356, 1:1000). The following day, the membranes were washed and then incubated with horseradish peroxidase-labeled anti-mouse or anti-rabbit secondary antibodies for 1 h at RT. Subsequently, membrane-bound horseradish peroxidase-labeled antibodies were detected using an enhanced chemiluminescence detection system including the Pierce ECL Western Blotting Substrate (Thermo Fisher Scientific). Quantification of the bands was performed by ImageJ 1.44 (Image Processing and Analysis in Java).

## Metabolite extraction

For lipidomics analysis, organic solvent extraction for two microglia cell samples was processed as follows. 1.2 million microglia cells were homogenized with 200 μL of pre-chilled 80% methanol using pulsed tip sonicator on dry ice. For protein precipitation, 150 μL of pre-chilled acetone and 50 μL of PBS were added and vortex rigorously for 60 s, then centrifuge at $20,000 \times g$ for 30 min at 4 °C. Supernatant contains no protein was transferred to 2 mL glass tube and was vaporized in a vacuum concentrator with no heat. 100 μL of PBS, 150 μL of pre-chilled methanol, and 300 μL of pre-chilled chloroform were added and vortexed rigorously for 60 s then centrifuged at $2900 \times g$ for 10 min at 4 °C. The bottom chloroform fraction was transferred to new 2 mL glass tubes. 300 μL of pre-chilled chloroform was added to the remaining upper fraction and vortexed rigorously for 60 sec then centrifuged at $2900 \times g$ for 10 min at 4 °C. The bottom chloroform fraction was combined to the previous chloroform fraction collected and was vaporized in a vacuum concentrator with no heat. Organic solvent extracted samples were reconstituted with 50 μL of 50% methanol.

## LC-MS analysis

Liquid chromatography separation was done with Waters BEH C8 column (2.1 mm × 100 mm, 1.7 um) using16 min gradient. Mobile phase A is 10 mM ammonium acetate with 5% methanol, 0.1% acetic acid in $H_2O$ and mobile phase B is 0.1% acetic acid in methanol. Flowrate was set at 0.4 mL/min and column oven temperature at 30 °C. 5 μL of samples were injected into Vanquish UHPLC (ThermoScientific) connected with Orbitrap ID-X Tribrid Mass Spectrometer (ThermoScientific). For global untargeted lipidomics, AcquireX DeepScan was applied and data was acquired in both negative and positive ion mode. The MS resolution was set at 60,000 for both MS1 and MS2 scans with scan range of 200–1100 $m/z$. The capillary voltage in the positive mode was 3.4 kV and the negative mode was 2.4 kV, respectively. The ion transfer tube temperature is 325 °C, Database search was performed by Compound Discoverer 3.3 with mzVault (MONA, GNPS, and NIST), mzCloud, and LipidBlast.

## Proteomics and data analysis

The proteins in Tau-Brain Extract (TBE) were precipitated by Acetone-TCA and the protein pellet was dissolved with 8 M Urea-containing buffer to subject for in-solution digestion. The sample was reduced with 4 mM dithiothreitol, and cysteines were alkylated with 15 mM

iodoacetamide. The proteins were digested with adding trypsin (1:50 trypsin to protein ratio) for overnight incubation at 37 °C. The tryptic peptide samples were desalted with SepPak C18 SPE cartridge and then subject to the SpeedVac dry. After lyophilization, peptides were reconstituted with 0.1% FA in water. Peptides were injected onto a Neo trap cartridge coupled with an analytical column (75 μm ID × 50 cm PepMap™ Neo C18, 2 μm). Samples were separated using a 2 hour of linear gradient of solvent A (0.1% formic acid in water) and solvent B (0.1% formic acid in ACN) using a Vanquish Neo UHPLC System coupled to an Orbitrap Exploris 480 Mass Spectrometer (Thermo Fisher Scientific).

The resulting tandem MS data was queried for protein identification against the *Mus musculus* database using Mascot v.2.8.0 (Matrix Science). The following modifications were set as search parameters: peptide mass tolerance at 10 ppm, trypsin enzyme, 3 allowed missed cleavage sites, carbamidomethylated cysteine (static modification), and oxidized methionine, deaminated asparagine/glutamine, and protein N-term acetylation (variable modification). The search results were validated with 1% FDR of protein threshold and 90% of peptide threshold using Scaffold v5.2.1 (Proteome Software).

### RNA sequencing and analysis

RNA sequencing and analysis were performed by the Genome Technology Access Center at Washington University using their standard methods, which are summarized here. Sample RNA integrity was determined using a Tapestation and library preparation was performed with 10 ng of total RNA for samples with a Bioanalyzer RIN score greater than 8.0. ds-cDNA was prepared using the SMARTer Ultra Low RNA kit for Illumina Sequencing (Takara-Clontech) per manufacturer's protocol. cDNA was then fragmented using a Covaris E220 sonicator using peak incident power 18, duty factor 20%, cycles per burst 50 for 120 s. The cDNA was blunt-ended, had an A base added to the 3′ ends, and then had Illumina sequencing adapters ligated to the ends. Ligated fragments were then amplified for 12–15 cycles using primers incorporating unique dual index tags. The fragments for each sample were then pooled in an equimolar ratio and sequenced on an Illumina NovaSeq-6000 using 150 base pair paired-end reads. Basecalls and demultiplexing were performed with Illumina's RTA 1.9 software and the reads were aligned to the *Mus musculus* Ensembl release 76 GRCm38 primary assembly with STAR version 2.5.1a. Gene counts were quantitated with Subread:featureCount version 1.4.6-p5. All gene counts were then imported into the R/Bioconductor package EdgeR and TMM normalization size factors were calculated to adjust the samples for differences in library size. Ribosomal genes were removed and only genes expressed greater than one count-per-million in at least four samples were kept for further analysis. The adjusted TMM size factors and the matrix of counts were then imported into the R/Bioconductor package Limma. Weighted likelihoods were then calculated for all samples and the count matrix was transformed to moderated log 2 counts-per-million with Limma's voom With Quality Weights. Differential expression analysis was then performed to analyze for differences between conditions and the results were filtered for only those genes with Benjamini–Hochberg false-discovery rate adjusted *p* values ≤0.05.

Single nucleus RNA-seq data shown in Fig. 5b, c was obtained from the Seattle Alzheimer's Disease Brain Cell Atlas (SEA-AD) consortium website (https://portal.brain-map.org/explore/seattle-alzheimers-disease) using the Transcriptomics Comparative Viewer.

### Statistical analysis

Sample size for mouse cohorts was determined based on sample size calculations using tau burden (AT8 staining % area) in the hippocampus as a primary endpoint, using previous data from the lab. Sample size calculation was designed to detect a 50% change in AT8

with 80% power and alpha = 0.05. All samples sizes were described in each figure legends. The mice and cultured cells are naturally randomized based on sex and genotype. Third-party animal staff with no knowledge of the mouse genotypes performed all mouse husbandry. During data preparation and analysis, mice are assigned a number, and genotype is not indicated.

For the statistical analysis, Student's *t* tests, or one- or two-way analyses of variance (ANOVAs) with Tukey post hoc tests were performed using GraphPad Prism software 8. *F* tests were performed on data to determine whether the variances differed significantly, in which case appropriate non-parametric tests were used. For two-way ANOVAs, multiple comparison testing was only performed if the main effect was significant at $p < 0.05$. Outliers were identified using Grubbs' test and were excluded. Statistical tests were performed with GraphPad Prizm software, version 9.4.0. $p < 0.05$ was considered significant and was noted with asterisks indicating the *p* value: *$p < 0.05$, **<0.01, ***<0.005, ****<0.001.

### Reporting summary

Further information on research design is available in the Nature Portfolio Reporting Summary linked to this article.

## Data availability

All data supporting this study are publicly available. Lipidomic data can be found on Figshare: https://doi.org/10.6084/m9.figshare.23669112. The RNA-seq data generated in this study have been deposited in the Gene Expression Omnibus (GEO) under the accession code: GSE236875. We utilized Mus musculus Ensembl release 76 GRCm38 primary assembly for RNA-seq. Lipidomic analysis is also provided in this paper as a Source Data file and all other data are available upon request. Source data are provided as a Source Data file. The mass spectrometry proteomic data have been deposited to the ProteomeXchange Consortium via the PRIDE[48] partner repository with the dataset identifier PXD043877 or at: https://www.ebi.ac.uk/pride/archive/projects/PXD043877. Human transcriptomic data is utilized by the data from AlzData web server [http://www.alzdata.org/download1.php], and the mouse ChIP-seq data sets are available in the NIH GEO repository under accession number GSE148644 or at https://www.ncbi.nlm.nih.gov/geo/query/acc.cgi?acc=GSE148644. Source data are provided with this paper.

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

## Acknowledgements

This work was supported by the following grants: NIA grants R01AG063743 (E.S.M.), RF1AG061776 (E.S.M.), Cure Alzheimer's Fund Research Grant (E.S.M.), McDonnell Center for Cellular and Molecular Neurobiology Postdoctoral Fellowship (J.L.), and NIH grants RF1AG062077 (M.L.), R35NS097273 (M.L.), RF1AG062171 (M.L.), P01NS084974-01 (M.L.), and R01DK45586 (M.L.), RF1AG062077 (L.P.), 586 R35NS097273 (L.P.), RF1AG062171 (L.P.), P01NS084974-01 (L.P.). We thank the assistance of Ross Kossina and Dr. James Fitzpatrick at the Washington University Center for Cellular Imaging (WUCCI) in electron microscopy studies, which is supported by Washington University School of Medicine, The Children's Discovery Institute of University and

St. Louis Children's Hospital (CDI-CORE-2015-505 and CDI-CORE-2019-813) and the Foundation for Barnes-Jewish Hospital (3770). We also thank the Genome Technology Access Center at the McDonnell Genome Institute at Washington University School of Medicine for help with genomic analysis. The Center is partially supported by NCI Cancer Center Support Grant #P30 CA91842 to the Siteman Cancer Center from the National Center for Research Resources (NCRR), a component of the National Institutes of Health (NIH), and NIH Roadmap for Medical Research. Mass Spectrometry analyses were performed by the Mass Spectrometry Technology Access Center at McDonnell Genome Institute (MTAC@MGI) at Washington University School of Medicine. The Seattle Alzheimer's Disease Brain Cell Atlas (SEA-AD) consortium is supported by the National Institutes on Aging (NIA) grant U19AG060909. Lastly, we also thank Yang Shi (Chinese Institute for Brain Research), Chanung Wang (Regeneron Pharmaceuticals), and Michael Strickland in Dr. David Holtzman's lab at Washington University for protocols and reagent assistance.

## Author contributions

J.L. and E.S.M. planned the study and designed the experiments. E.S.M. provided funding support. J.L. performed and analyzed all experiments except for those indicated below. J.M.D performed LPS injection experiments. J.H.S., M.S., and Y.A.G. collected and analyzed lipidomic and proteomic data. M.W.K. provided bulk RNA-seq data from isolated microglia and P.W.S. assisted with sequencing analysis. G.T.T. assisted with RNA-use data acquisition. M.A.L. shared ChIP-seq data, provided REV-ERBα-floxed mice, and assisted with experiment design and data interpretation. L.P. provided the AAV-P301L virus and assisted with experiment design and data interpretation. J.L. and E.S.M. initially wrote the manuscript and revised it.

## Competing interests

The authors declare no competing interests.
