## [Peer Review File · Nature Communications]

Microglial REV-ERB α regulates inflammation and lipid droplet formation to drive tauopathy in male miceREVIEWER COMMENTS

Reviewer #1 (Remarks to the Author):

The manuscript “Microglial REV-ERB α regulates lipid droplet formation and tauopathy in a sex dependent manner” investigates the connection between REV-ERB α , PLIN2, lipid droplets, sex, and tauopathy. The researchers demonstrate that REV-ERB α regulates PLIN2, a lipid droplet associated protein and that this regulation of PLIN2 regulates lipid droplet formation. They then suggest that lipid droplet formation (LD) controls tau uptake and that this control of tau uptake is sex specific. The work then pivots to the sex specific effect of the inflammatory response to tau in a REV-ERB α KO, which appears to be regulated differently depending on the sex of the organism. The authors conclude that the connection they show between lipid droplet formation and the clock is another reason the clock is a good therapeutic target for ALZ treatment. The link between all these elements is very interesting, as sex, the clock, immunity, lipid droplet formation, and ALZ are all relevant and exciting areas of study.

However, in the overall picture, there are issues with both the research and the writing of this manuscript. First, the manuscript seems to be a summary of two different research topics that have been combined and the links between each separate study are not well parsed. Do the authors intend this to be a study about how Tau uptake is regulating the clock in a sex dependent manner OR in a LD dependent manner? Figures 5-8 seems to reference the second story and figures 1-4 the first. A connecting experiment, like regulating lipid droplets in vivo in the REV-ERB α strains would go a long way to bring this work from two stories into one. Along with this, there are some directionality issues with the experimental findings. For example, is PLIN2 repression by REV-ERB α controlling LD or is LD controlling PLIN2 levels? Both are shown by this manuscript but how these things could both happen at the same time and what it means is not investigated. Additionally, there are some holes in the research that the authors present. For example, there is very little evidence presented in this manuscript that LD is controlling tau uptake, so how can the authors conclude that there is a relationship between PLIN2 and tau uptake? Also, there is no PLIN2 protein to impart the rhythm that NR1D1 regulates in PLIN2 mRNA. Finally, there are a lot of clarity issues that need to be addressed before an assessment of the findings can be appropriately completed.

Major notes:

- 1) The authors note that their data shows that NR1D1 binds to the PLIN2 promoter and represses PLIN2 expression. While the data presented is clear in the binding of NR1D1 to the promoter, that it in turn represses PLIN2 is not shown. While it is a reasonable assumption that in this case NR1D1 represses its target, as it does for most targets, repressors can often act as activators as well and this should be shown if it is going to be asserted.
- 2) The authors assert that PLIN2 and 3 are only minimally expressed in neurons. However, their CHIP data and data about expression levels being higher in AD vs control patients would suggest that there is expression in neuronal populations. Explaining this discordance is important.
- 3) The authors refer to a citation that is currently “in preparation” to talk about data related to translomics. However, this data is not described in the methods and the manuscript is not accessible online, making it impossible to review. Therefore, this data is not appropriate to add to the manuscript.
- 4) The authors show that PLIN2 mRNA is circadianly timed antiphase to NR1D1 using transcriptomic data from “adult mice”. However, they then go on to show that at the protein level, there is little to no PLIN2 in young or old mice, only in NR1D1 kos. This begs the question, what is the relevance of the rhythm in transcript level of PLIN2 if no protein is seen in mice that are old or young? How does NR1D1 impart a rhythm on the LD formation via PLIN2 if there is no protein present?
- 5) The authors note that they see male and female differences in the level of the response to LPS. However, this could be a circadian effect, as TNF α and IL6 are known to be circadian in some tissues, and the phase of their rhythms could be sex specific. So, the levels could be peaking at different times and the authors did not look in the correct phase. This should be addressed.
- 6) The authors note that when they added a922500, which inhibited dgat1 activity, it decreased both LD accumulation, as expected, as well as PLIN2 expression. This seems backwards, as the authors are arguing that the levels of NR1D1 regulates PLIN2 expression which in turn effects LD formation. However, this result suggests that LD formation regulated PLIN2 instead. How do the authors rationalize this? This is also noted in figure 4.
- 7) To be clear, was the data from figure 2c and f performed in males or females? Because if it was females, it contradicts the data from 2 d-e.

- 8) A similar question for figure 3, was this done in males or in females or in both?
- 9) In figure 3, there appears to be no additive effect of tbe and NR1D1 ko cells. This suggests that the knockout effects of NR1D1 are epistatic to the inflammation that is induced by the addition of Tau. It begs the question, instead of a regulation of PLIN2, could all of the phenotypes seen relate back to the effect of increased inflammation that suppresses clock function in immune cells, as has been shown by several labs in the circadian field?
- 10) The authors used OA to increase LD levels rather than using TBE. What was the reason for this and can the OA levels actually mirror the TBE effects?
- 11) The data supporting that LD is controlling Tau uptake is very weak in this manuscript, with almost no rescue of uptake with LD inhibitors in the TBE model. How can the authors state with certainty that this supports their theory. It seems to in fact suggest that there is another mechanism. The authors current hypothesis is that this means that the effects of NR1D1 come earlier in the disease but there is no evidence that they present to rationalize the idea that NR1D1 would play a role early on in the disease phenotype.
- 12) In the section referring to figure 5, the authors claim to have looked at IBA1, but where, how, and when they did this is not shown in the figures.
- 13) The authors note that females have no increase in immune pathways but did not acknowledge or discuss the fact that C1q is up in both males and females and should do so.

Minor notes:

- 1) The authors should review for the many minor grammatical errors in the document (e.g., line 45 "Recent studies show that lipid homeostasis and lipid droplet (LD) formation is disrupted in microglia in AD models and LD formation in microglial has been highlighted as a potential therapeutic target⁵. Lipid-droplet accumulating microglia (LDAM) have been observed in mouse and human brain during ageing and are an emerging pathological phenotype in AD⁶).
- 2) Some of the predicted and discussed relationships between the data are hard to follow and could benefit from pictograms to describe it.
- 3) The discussion section spends a great deal of time rehashing the results and should focus more on discussing them.
- 4) In general, this manuscript is very complex, with data that at times seems unrelated to other data. Is this a story about males vs females, the circadian clock, inflammation, etc.?

The connections between each idea could be more strongly supported and suggest that there are possibly more than one manuscript worth of data in this paper.

5) In extended data figure 3c, the authors note that they looked at PLIN2, Il6, Tnfa, and Il1b, but Il6 was not labeled in the figure. Please make sure it is labeled.

Reviewer #2 (Remarks to the Author):

This study focuses on the role of REV-ERB α in the modulation of lipid droplets, microglia activity and subsequent mediation of tau pathology in a sex dependent manner. The authors utilized REV-ERB α KO mice to identify a sex dependent role of REV-ERB α in mediation of microglia response to tau induced stress and tau phagocytosis. Further, they crossed microglia specific REV-ERB α KO mice with the P301S tau transgenic mice and injected AAV-mediated tau to the microglial REV-ERB α KO mice. The authors show that REV-ERB α loss-of-function worsens tau pathology in male but not in female mice. The premise of the study is intriguing and novel in the attempt to identify a sex-specific role of microglial REV-ERB α in tau pathogenesis. However, additional studies are required to strengthen their conclusions. In particular, the claim that Plin2 is a direct transcriptional target of REV-ERB α needs to be further substantiated. The sex specific effects, while interesting, are under-developed. The quality of certain data need to be improved. In addition, the authors reported that germline REV-ERB α KO reduced Ab pathology (PMCID: 6996949) but microglial REV-ERB α KO increased tau pathology. The seemingly opposite effects need to be thoroughly discussed. The following are the specific concerns:

- Figure 1

- o It is not clear why microarray, instead of scRNAseq data, was used to examine REV-ERB α expression in human AD as scRNAseq provides much deeper insights into cell-type and sex differences as the authors indicated (line 58-81). The human and mouse single cell datasets should be used to identify possible cell-type (microglia) and sex-specific differences of Nr1d1 (REV-ERB α) and PLIN2 in controls and in AD samples.

- o To demonstrate a direct transcriptional regulation of Plin2 by REV-ERB α , luciferase activity assay using the Plin2 WT and REV-ERB α binding deficient promoters should be performed.

- o It is rather strange that APP/PS1 samples were used to characterize Plin2 expression in microglia and astrocyte. Since the study is centered on the role of REV-ERB α on tau

handling, the experiment needs to be done in P301S mice.

o Fig. 1f, suggest including PLIN2 staining in young REV-ERB α KO. In addition, add DAPI in the corresponding images to document comparable hippocampal regions between groups.

Is increased PLIN2 sex dependent?

- Figure 2

o It is intriguing that REV-ERB α KO increased BODIPY+ LDs in both male and female microglial cultures but that only male microglia showed increased BODIPY+ LDs when treated with Tau-enriched Brain Extract (TBE). Is this sex-dependent effect TBE specific or a general phenomenon? How about LPS induced LDs? Are the changes in LDs associated with differences in Plin2 levels?

o It appears that addition of TBE induces LDs as well (compare 2a and 2d, Cre-). The effect of TBE should be quantified. Is this sex dependent?

o Fig. 2c: the amount of TBE should be specified.

o Fig. 2f: x-axis labels are mis-aligned.

- Figure 4

o Fig. 4e: Adding the FITC-tau uptake in just the iDGAT1 treated Control and RKO cells.

o Include inflammation cytokines previously examined in Fig4a in the conditions used in Fig4e.

- Figure 5

o Iba1 staining is mentioned in text for figure 5c but representative images and analysis are not shown in the figure.

o Include astrocyte (GFAP) and microglia (Aif1) staining in LPS treated REV-ERB α microglia KO (Fig 5e). Are changes between Cre- and Cre+ males significant?

- Figures 6 and 7

o The two figures are the same except male and female mice were analyzed. To better appreciate the sex-differences, the two figures should be combined and the displayed and quantified side-by-side (similar to Fig. 8).

o Higher resolution staining images (similar to Fig. 8b and c) need to be provided to properly evaluate the quality of the data.

o REV-ERB KO in microglia should be validated by co-staining for REV-ERB α with Iba1 or by qPCR of sorted microglia.

Minor comments/suggestions:

- There are a number of typos throughout the manuscript, e.g. “REV-ERB α causes induction of PLIN2 and LDs...” (line 1480149) should be REV-ERB α KO.
- Remove representative flow cytometry plots to supplementary.
- It is puzzling that 18mon P301S tau mutant mouse brain was used to produce “Tau-enriched Brain Extract (TBE)” as these mice are known to develop hind limb paralysis by 9-12 months. Can authors comment on the phenotypes of the P301S mice used in their study?

Reviewer #3 (Remarks to the Author):

The key findings from this study suggest a strong sex-dependent link between lipid droplet (LD) accumulation in microglia and an upregulation of the housekeeping protein PLIN 2 on LD surfaces. The authors promote the idea that a circadian suppressor REV-ERB α deletion leads to microglia Plin2 accumulation in aged brain and that this is amplified by tau mediated inflammation. Activation of microglia with inflammogen LPS is strongly enhanced in male but not in female mice following REV-ERB α deletion. REV-ERB α deletion exacerbates pathology specifically in male P301S tau transgenic mice as assessed by IHC of neural cells and phagocytosis in microglia. Taken together the data suggest that REV-ERB α is a potential therapeutic target as a sex dependent regulator of microglial lipid metabolism in tauopathies.

These findings reported in the Communication are noteworthy, but data on lipid species and electrophysiology could make the claims stronger and open new avenues in therapeutic approaches in tauopathies. Missing the functional analyses (except phagocytosis) raises some concerns. In addition, concluding statement (abstract) that data demonstrate rather than suggest is a little exaggerated considering that no lipidomic analysis for neutral and other lipids was performed. However, findings are important and changes in microglia shown on increasingly complex models from in vitro to in vivo and limited but relevant human AD brain samples make these studies attractive.

1. The value of the work and its potential significance to the field is mainly in its logical combination of leads from numerous studies in the field which provided some stepping stones in biology of microglia and sex difference, lipid droplets and tau research. Thus, it is somewhat limited in novelty, but it is valuable as a study combining sex-dependent LD changes in microglia and the effects of circadian suppressor REV-ERB α on LD abundance and

Plin 2 expression. Sex dependence on LD changes is very convincing and highlights a greater susceptibility to tau pathology in males. A partial restoration of LD homeostasis by manipulation of REV-ERBa suggests a participation of other genes aside from REV-ERBa and these candidate genes are not indicated.

2. The work provides relevant information from the current literature in the introduction, but lacks some of them in the discussion. Authors provide a link with earlier studies on LPS as a commonly used proinflammatory agent and compare it with uncharacterized cell culture media both of which produce sex-dependent effects. This part of the study is interesting but, the composition of this media is unknown and the main players are not suggested. New and important references on lipid droplet properties and composition should be included.

3. The work supports the major claims but there are some limitations. For instance, the only functional assay is phagocytosis of tau in microglia. Although it is an important assessment of microglia function it should be linked with neuronal functions by using electrophysiological techniques. Electrophysiological assessment of male and female REV-ERBa KO crossed with triple transgenic mice would be more convincing that the enhancement of LD abundance impairs neuronal functions. Cuervo et al., previously suggested a link between LD abundance in neurons and lipid transfer from neurons to glia. If the accumulation of LD is mainly in microglia due to the REV-ERBa KO but does not bear marked functional neuronal impairment the significance of the study is somewhat limited. Thus additional evidence for functional dependence of microglia LD on neuronal function is needed.

4. Are there flaws in the data analysis or interpretation?

There are no major flaws detected which would prohibit publication or require revisions. Additional experiments are needed: (i) media used to show the induction of LD should be characterized (lipidomic and proteomic analysis is required to suggest at least few possible relevant players), (ii) lipidomic analysis of LD should be performed and (iii) electrophysiology to show the neuronal functions in REV-ERB KO should be conducted.

5. The methodology is sound and the work meets the expected standards in the field. There is no novelty in this section.

6. There is enough detail provided in the methods for the work to be reproduced but some justifications of selected times are needed. For instance: Why 7 days for phagocytosis of FITC-tau for EM and 2 hrs of FACS in vitro? OA-induced LD formation can be partly reduced

with pharmacological inhibitors of DGAT1 but this was not achieved with TBE suggesting that a complex “soup” of stimuli do not respond to these inhibitors and imply the contribution other than DGAT. This is interesting but could be discussed in more detail if the composition of the TBE medium is analyzed. Formation of LD by OA treatment is well established and serves here as one control but this is not a suitable control for TBE. At least some components of TBE must be identified and tested to show which of them induce LD formation and if these LD contain lipotoxins. It is not only the appearance of LDs but also their quality – i.e. lipid composition which will contribute to the functional impairment of neural cells. Thus, the following experiments are suggested: (i) determination of the major components in TBE which contribute to LD formation and their lipotoxicity; (ii) lipidomic analysis to determine the lipid species in microglia and neurons upon TBE stimulation. (iii) electrophysiological assessment of neuronal functions upon TBE stimulation in a sex-dependent manner.

7. The strongest and most convincing evidence in the entire study is a sex dependent effects in REV-ERBa KO in the tau mouse model. It would be interesting to see if another circadian rhythm and clock gene is involved in lipid regulation in the investigated animal model. This aspect could be addressed in the discussion.

REVIEWER COMMENTS

Reviewer #1 (Remarks to the Author):

The manuscript “Microglial REV-ERBa regulates lipid droplet formation and tauopathy in a sex dependent manner” investigates the connection between REV-ERBa, PLIN2, lipid droplets, sex, and tauopathy. The researchers demonstrate that REV-ERBa regulates PLIN2, a lipid droplet associated protein and that this regulation of PLIN2 regulates lipid droplet formation. They then suggest that lipid droplet formation (LD) controls tau uptake and that this control of tau uptake is sex specific. The work then pivots to the sex specific effect of the inflammatory response to tau in a REV-ERBa KO, which appears to be regulated differently depending on the sex of the organism. The authors conclude that the connection they show between lipid droplet formation and the clock is another reason the clock is a good therapeutic target for ALZ treatment. The link between all these elements is very interesting, as sex, the clock, immunity, lipid droplet formation, and ALZ are all relevant and exciting areas of study.

However, in the overall picture, there are issues with both the research and the writing of this manuscript. First, the manuscript seems to be a summary of two different research topics that have been combined and the links between each separate study are not well parsed. Do the authors intend this to be a study about how Tau uptake is regulating the clock in a sex dependent manner OR in a LD dependent manner? Figures 5-8 seems to reference the second story and figures 1-4 the first. A connecting experiment, like regulating lipid droplets in vivo in the REV-ERBa strains would go a long way to bring this work from two stories into one. Along with this, there are some directionality issues with the experimental findings. For example, is PLIN2 repression by REV-ERBa controlling LD or is LD controlling PLIN2 levels? Both are shown by this manuscript but how these things could both happen at the same time and what it means is not investigated. Additionally, there are some holes in the research that the authors present. For example, there is very little evidence presented in this manuscript that LD is controlling tau uptake, so how can the authors conclude that there is a relationship between PLIN2 and tau uptake? Also, there is no PLIN2 protein to impart the rhythm that NR1D1 regulates in PLIN2 mRNA. Finally, there are a lot of clarity issues that need to be addressed before an assessment of the findings can be appropriately completed.

We appreciate these comments from the reviewer and have made a concerted effort to bring the two halves of the story together. To do this, we have refocused the story on the idea that REV-ERB α deletion in microglia induces both inflammatory signaling as well as dysregulated lipid homeostasis, and that these two processes interact with each other to impair tau uptake and promote tau pathology in vivo. As we show below, PLIN2 appears to be more of a marker of LDs than a regulator of them, as we have now found that knocking down PLIN2 in microglia does not prevent LD formation. Thus, we have de-emphasized PLIN2 (though we still show that REV-ERB α can bind PLIN2 promoter and now show that it regulates mRNA levels), and now provide RNAseq data showing that REV-ERB α KO dysregulates multiple genes involved in lipid homeostasis in microglia, as well as inflammatory pathways.

Major notes:

1) The authors note that their data shows that NR1D1 binds to the PLIN2 promoter and represses PLIN2 expression. While the data presented is clear in the binding of NR1D1 to the promoter, that it in turn represses PLIN2 is not shown. While it is a reasonable assumption that in this case NR1D1 represses its target, as it does for most targets, repressors can often act as activators as well and this should be shown if it is going to be asserted.

- To address this point, we measured *Plin2* transcript in REV-ERB α KO microglia and observed that *Plin2* was significantly upregulated by REV-ERB α deletion. The data was shown at Fig. 4i.

2) The authors assert that PLIN2 and 3 are only minimally expressed in neurons. However, their CHIP data and data about expression levels being higher in AD vs control patients would suggest that there is expression in neuronal populations. Explaining this discordance is important.

- This is an important observation. To address this, we have now added snRNAseq data from the Seattle Alzheimer's Disease Brain Cell Atlas (SEA-AD) which supports *Plin2* expression in AD patient brain particularly in microglia. This data is now presented in Fig. 5b-c.

3) The authors refer to a citation that is currently “in preparation” to talk about data related to translationalomics. However, this data is not described in the methods and the manuscript is not accessible online, making it impossible to review. Therefore, this data is not appropriate to add to the manuscript.

- We removed the contents and citation from our manuscript because that paper is still not published yet.

4) The authors show that PLIN2 mRNA is circadianly timed antiphase to NR1D1 using transcriptomic data from “adult mice”. However, they then go on to show that at the protein level, there is little to no PLIN2 in young or old mice, only in NR1D1 *kos*. This begs the question, what is the relevance of the rhythm in transcript level of PLIN2 if no protein is seen in mice that are old or young? How does NR1D1 impart a rhythm on the LD formation via PLIN2 if there is no protein present?

- We do think PLIN2 is expressed in young mice, but the signal was very weak. Some PLIN2 positive green signal existed on young mice brain sections, but it is difficult to discern from background. In general, bright PLIN2 signal is only seen when there is significant LD accumulation in cells in the setting of disease (such as in aged ApoE4-P301S tau mice). To simplify this figure, we now present only aged WT and RKO mouse brain in Fig. 5h. Regarding rhythm of Nr1d1/PLIN2 expression, we don’t think much can be said about circadian rhythm based on 2 timepoints. Rather, we intended to show that low Nr1d1 levels were associated with higher Plin2 expression levels (at the RNA level), just to be consistent with our results showing that Nr1d1 KO causes increased PLIN2 expression.

5) The authors note that they see male and female differences in the level of the response to LPS. However, this could be a circadian effect, as TNF α and IL6 are known to be circadian in some tissues, and the phase of their rhythms could be sex specific. So, the levels could be peaking at different times and the authors did not look in the correct phase. This should be addressed.

- As we shown at our previous work (Griffin *et al.*, 2019), microglial activation varies by time of day, but REV-ERB α KO abrogates this time of day effect, locking the inflammatory response at a maximal state at both timepoints. Thus, we would expect LPS response in microglial REV-ERB α KO mice to be maximally saturated at both timepoints, and we assume the trend and the results will not be different between AM and PM.

6) The authors note that when they added a922500, which inhibited dgat1 activity, it decreased both LD accumulation, as expected, as well as PLIN2 expression. This seems backwards, as the authors are arguing that the levels of NR1D1 regulates PLIN2 expression which in turn effects LD formation. However, this result suggests that LD formation regulated PLIN2 instead. How do the authors rationalize this? This is also noted in figure 4.

- This is an excellent point, and we have changed our model based on further investigation of this issue. To test whether PLIN2 is essential for LD formation and microglial activity for tau uptake, we did both BODIPY and tau uptake assays using PLIN2 siRNA-treated cultured microglia (PLIN2 KD). However, PLIN2 KD did not alter LDs expression or microglial internalization of tau (see new Fig. S5). From this result, we suspect that despite our observation that REV-ERB α may regulate PLIN2, PLIN2 does not appear to mediate the effects REV-ERB α KO on LDs. We have changed the focus to look more broadly at regulation of microglial lipids by REV-ERB α as a more important mechanism and promising strategy for rescuing of microglial tau uptake. To investigate whether lipid metabolism is affected by REV-ERB α deletion, we performed RNA-seq using cultured WT control and REV-ERB α KO microglia. From the RNA-seq results, we found that general lipid processes are upregulated by REV-ERB α KO (now presented in Fig. 4b).

b

Table TOP10 Biological Processes				
GO Term	Biological Process	p-val	Fold enrich	FDR
GO:0071346	Cellular response to interferon-gamma	2.71E-07	5.37	8.7E-04
GO:0006935	Chemotaxis	2.15E-05	4.04	3.4E-02
GO:0001775	Cell activation	3.54E-05	14.64	3.8E-02
GO:0002376	Immune system process	5.37E-05	2.23	4.3E-02
GO:0006606	Protein import into nucleus	2.09E-04	3.99	1.1E-01
GO:0032760	Positive regulation of tumor necrosis factor production	3.86E-04	3.72	1.6E-01
GO:0006629	Lipid metabolic process	4.53E-04	1.86	1.6E-01
GO:0009636	Response to toxic substance	7.76E-04	4.07	2.1E-01
GO:0006397	mRNA processing	8.61E-04	2.26	2.1E-01
GO:0006898	Receptor-mediated endocytosis	9.01E-04	4.45	2.1E-01

7) To be clear, was the data from figure 2c and f performed in males or females? Because if it was females, it contradicts the data from 2 d-e.

- We used mixed sex WT microglia for 2c and used male microglia for 2f (Those figures are moved at Fig. S2d and S3c respectively in revised manuscript). Increasing BODIPY levels on REV-ERB α KO microglia is common phenomenon on both male and female at basal levels, but only males show exacerbation in response to tau (TBE).

8) A similar question for figure 3, was this done in males or in females or in both?

- Mixed sex WT microglia is used to test the effect of TBE on tau uptake assay (Fig. 3c and 3d) and we used male microglia for Fig. 3e.

9) In figure 3, there appears to be no additive effect of TBE and NR1D1 ko cells. This suggests that the knockout effects of NR1D1 are epistatic to the inflammation that is induced by the addition of Tau. It begs the question, instead of a regulation of PLIN2, could all of the phenotypes seen relate back to the effect of increased inflammation that suppresses clock function in immune cells, as has been shown by several labs in the circadian field?

- This is a helpful comment, and we have refocused some of the paper to address this. As mentioned above, we no longer argue that PLIN2 mediates the effects of REV-ERB α KO directly, but that REV-ERB α regulates multiple lipid metabolic pathways. We also address the idea that inflammation (which is augmented in REV-ERB α KO microglia in Fig. 1) may drive some of the lipid droplet phenotype. We show in new Fig. 4d,e that LPS can induce LDs in microglia, and in Fig. S4 that LPS can also reduce tau uptake by microglia. Since we observe increased inflammation, LDs, and impaired tau uptake in REV-ERB α KO microglia, we now argue that the increase in both inflammation and lipid metabolism may combine to suppress tau uptake.

As reviewer mentioned in this question, TBE induces massive inflammation, and the increase in LDs is not blocked by iDGAT drugs (new Fig. S3d). For this reason, we treated REV-ERB α KO microglia with LD blockers without TBE and observed recovery of tau uptake. The data is now presented at Fig. 6i (and shown to the right). This difference may be due to the fact that REV-ERB α KO does not cause nearly as much inflammation as TBE. As an extension of this result, we suspect that in late phase of AD which shows saturated inflammation and severe metabolic changes, is probably not suitable to therapy for targeting lipid regulation to enhance tau phagocytosis. We discussed the timing of REV-ERB α -targeted treatments for AD in the discussion.

10) The authors used OA to increase LD levels rather than using TBE. What was the reason for this and can the OA levels actually mirror the TBE effects?

- Although TBE evokes LDs accumulation in microglia, it reflects tau-treated condition with diverse metabolic changes, but not pure lipid treated condition. In this reason, OA used as a representative means of specifically increasing lipid content. Our data now shows that lipid-treated or lipid containing microglia show weak tau phagocytic activity. We do not think OA mirrors the effect of TBE perfectly, but it supports the importance of lipid droplets on microglial function for tau and draws a parallel with lipid changes occurring in REV-ERB α KO microglia.

11) The data supporting that LD is controlling Tau uptake is very weak in this manuscript, with almost no rescue of uptake with LD inhibitors in the TBE model. How can the authors state with certainty that this supports their theory. It seems to in fact suggest that there is another mechanism. The authors current hypothesis is that this means that the effects of NR1D1 come earlier in the disease but there is no evidence that they present to rationalize the idea that NR1D1 would play a role early on in the disease phenotype.

We agree that the inability of LD inhibitors to rescue tau uptake in TBE-treated microglia is an issue. We think that TBE may saturate the cells with tau, cause a competitive inhibition of uptake. As mentioned above, TBE also

induces severe inflammation, which may inhibit tau uptake itself (as we see with LPS-treated microglia in Fig. S4). Thus, we have moved away from using TBE a bit in the revision (most of the data is now in Supplement), and emphasize data that LD inhibitors can partially rescue tau uptake in OA-treated microglia. We also have performed both BODIPY and tau uptake assay using iDGAT1-treated REV-ERB α KO microglia without TBE and observed some rescue of tau uptake (Fig. 6i). Our data consistently show that LD inhibitors exert a partial (modest) effect on tau uptake, though it may not be the primary driver. We agree that inflammation also plays a role, and have integrated that concept into this revision. These data also support the hypothesis that lipid modulation by REV-ERB α should be considered at the early phase of AD, before severe tau pathology is present.

12) In the section referring to figure 5, the authors claim to have looked at IBA1, but where, how, and when they did this is not shown in the figures.

- Thank you for noticing this oversight. We updated IBA1 data in Fig. 1d and 1e on revised manuscript. We apologize.

13) The authors note that females have no increase in immune pathways but did not acknowledge or discuss the fact that C1q is up in both males and females and should do so.

- We measured C1q mRNA levels in P301S mice in Fig. 2, and show that C1q increases in male Cre⁺ mice (Fig. 2d) and decreases in female Cre⁺ mice (Fig. 2e).

Minor notes:

1) The authors should review for the many minor grammatical errors in the document (e.g., line 45 “Recent studies show that lipid homeostasis and lipid droplet (LD) formation is disrupted in microglia in AD models and LD formation in microglial has been highlighted as a potential therapeutic target⁵. Lipid-droplet accumulating microglia (LDAM) have been observed in mouse and human brain during ageing and are an emerging pathological phenotype in AD⁶.”).

- We have edited the manuscript thoroughly and hopefully caught most of these errors.

2) Some of the predicted and discussed relationships between the data are hard to follow and could benefit from pictograms to describe it.

- We generated graphic summary at Fig. 8 and hope it will be helpful to understand.

3) The discussion section spends a great deal of time rehashing the results and should focus more on discussing them.

- We re-generated the 'Discussion' part with additional viewpoints.

4) In general, this manuscript is very complex, with data that at times seems unrelated to other data. Is this

a story about males vs females, the circadian clock, inflammation, etc.? The connections between each idea could be more strongly supported and suggest that there are possibly more than one manuscript worth of data in this paper.

- We have refocused the revised manuscript to address the idea that REV-ERB α KO causes inflammation and lipid droplet formation in microglia, impairing tau phagocytosis and increasing tau pathology. While we observe that this occurs primarily in male mice, we think this may be a product of the profound inflammatory difference between male and female microglia, and we do not pursue the underlying mechanisms of the sex difference in this paper.

5) In extended data figure 3c, the authors note that they looked at PLIN2, Il6, Tnf α , and Il1b, but Il6 was not labeled in the figure. Please make sure it is labeled.

- Those graphs are moved to Fig. S2e, and labeled with correct gene names.

Reviewer #2 (Remarks to the Author):

This study focuses on the role of REV-ERB α in the modulation of lipid droplets, microglia activity and subsequent mediation of tau pathology in a sex dependent manner. The authors utilized REV-ERB α KO mice to identify a sex dependent role of REV-ERB α in mediation of microglia response to tau induced stress and tau phagocytosis. Further, they crossed microglia specific REV-ERB α KO mice with the P301S tau transgenic mice and injected AAV-mediated tau to the microglial REV-ERB α KO mice. The authors show that REV-ERB α loss-of-function worsens tau pathology in male but not in female mice. The premise of the study is intriguing

and novel in the attempt to identify a sex-specific role of microglial REV-ERB α in tau pathogenesis. However, additional studies are required to strengthen their conclusions. In particular, the claim that Plin2 is a direct transcriptional target of REV-ERB α needs to be further substantiated. The sex specific effects, while interesting, are under-developed. The quality of certain data need to be improved. In addition, the authors reported that germline REV-ERB α KO reduced Ab pathology (PMCID: 6996949) but microglial REV-ERB α KO increased tau pathology. The seemingly opposite effects need to be thoroughly discussed. The following are the specific concerns:

Thank you for these astute comments. As mentioned above, we have now changed our stance on PLIN2 as a mediator of the effects of REV-ERB α , and have re-written most of the paper and added more data. We also address the discrepancy between our previous publication on A β phagocytosis and our current data on tau phagocytosis, as REV-ERB α seems to have opposite effects on these processes.

• **Figure 1**

1) It is not clear why microarray, instead of scRNAseq data, was used to examine REV-ERB α expression in human AD as scRNAseq provides much deeper insights into cell-type and sex differences as the authors indicated (line 58-81). The human and mouse single cell datasets should be used to identify possible cell-type (microglia) and sex-specific differences of Nr1d1 (REV-ERB α) and PLIN2 in controls and in AD samples.

- To address this question, we have now added snRNAseq data from the Seattle Alzheimer's Disease Brain Cell Atlas (SEA-AD) and found that Plin2 was primarily expressed in microglia. This data is presented in Fig. 5b-c. On the other hand, Rev-erb α (Nr1d1) is broadly expressed throughout the brain (This graph is only for rebuttal letter).

With the available data, we were not able to address sex-dependent effects in microglia.

2) To demonstrate a direct transcriptional regulation of Plin2 by REV-ERB α , luciferase activity assay using the Plin2 WT and REV-ERB α binding deficient promoters should be performed.

- In general, primary microglia are hard to transfect the plasmid DNA for overexpression and present super low efficiency. Instead of a promoter assay, we confirmed that *Plin2* transcript is upregulated by REV-ERB α KD in microglia and the data is shown at Fig. 1c.

3) It is rather strange that APP/PS1 samples were used to characterize Plin2 expression in microglia and astrocyte. Since the study is centered on the role of REV-ERB α on tau handling, the experiment needs to be done in P301S mice.

We agree and have removed this data from APP/PS1 mice. We found that previous report showed that Plin2 significantly increased in isolated microglia from P301S tau transgenic mice compared to control group. We utilized their open source and see the table like as below. Moreover, other supportive studies also suggest high levels of Plin2 in tau transgenic mice (Srinivasan K *et al.*, 2020, Patel T *et al.*, 2022.)

i Cultured microglia

Supplementary Table 6: DE mRNAs between PS19 vs Control male microglia					
n = 5 Ctrl samples, 3 PS19 samples, 2 mice/sample. Wald test used. P value correction by Benjamini-Hochberg.					
	baseMean	log2FoldChange (PS vs Ctrl)	pvalue	padj	gene
ENSMUSG00000028494	267.613013	1.286781501	0.0006739	0.035470274	Plin2

4) Fig. 1f, suggest including PLIN2 staining in young REV-ERB α KO. In addition, add DAPI in the corresponding images to document comparable hippocampal regions between groups. Is increased PLIN2 sex dependent?

- Unfortunately, we do not have young REV-ERB α KO brain sections, and no longer keep the germline REV-ERB α KO mouse line. However, we generally are not able to detect PLIN2 staining via IHC unless there is very significant pathology (such as in end-stage tau-related neurodegeneration) so it's unlikely we would see much. We have shown the PLIN2 images to the right, but over-exposed. In this way, the CA3 region is clear and shows slight PLIN2 positivity in aged WT mice. The CA3 region is very evident under the microscope simply by increasing the brightness, which is why DAPI was not included. We do not have enough samples to do proper sex stratification in this experiment (as we used very old mice that were still in our colony, but are now gone). However, as we shown at Fig. 4g, since male and female REV-ERB α KO microglia did not showed sex-differences on LDs accumulation under the basal levels, we suspect that *in vivo* will be the same. We re-presented it at here.

• **Figure 2**

5) It is intriguing that REV-ERB α KO increased BODIPY+ LDs in both male and female microglial cultures but that only male microglia showed increased BODIPY+ LDs when treated with Tau-enriched Brain Extract (TBE). Is this sex-dependent effect TBE specific or a general phenomenon? How about LPS induced LDs? Are the changes in LDs associated with differences in Plin2 levels?

- Since we observed sex-dependent effect of REV-ERB α on tau pathology and LPS-injected responses between male and female mice (and in cells), we assume the sex-differences are general phenomena.

- To test whether LPS also evoked LDs and microglial tau phagocytic activity, we performed both BODIPY staining and tau uptake assay. LPS has a similar effect as REV-ERB α KO or TBE treatment on cultured microglia, as BODIPY+ cells were increased (Fig. 4e) and the tau uptake by microglia was decreased in LPS-treated cells (Fig. S4). Interestingly, Plin2 transcript was dramatically downregulated in LPS-treated microglia, unlike TBE-treated, suggesting that the change in LDs was not correlated with *Plin2*.

6) It appears that addition of TBE induces LDs as well (compare 2a and 2d, Cre-). The effect of TBE should be quantified. Is this sex dependent?

- We added statistical analysis in this graph and confirmed that LDs are not different between Cre- male and Cre+ male at basal levels. Male and female data now shown in Fig. 4f and 4g.

7) Fig. 2c: the amount of TBE should be specified

- In Fig. 2c, cells were treated with 2, 4, and 8 μl of TBE. We quantified pTau levels on TBE using sandwich ELISA method and verified that pTau existed in TBE with $2.7\mu\text{g}/\mu\text{l}$ concentration. The data is presented at Fig. S2 c.

8) Fig. 2f: x-axis labels are mis-aligned.

- We fixed this and moved this graph to Fig. S3c.

• **Figure 4**

9) Fig. 4e: Adding the FITC-tau uptake in just the iDGAT1 treated Control and RKO cells.

- To address it, we treated Control (Cre-) and RKO (Cre+) microglia with iDGAT1 and performed BODIPY staining and tau uptake assay. iDGAT1 treatment did not fully suppress LD formation in REV-ERB α KO microglia, which was disappointing. However, despite the very modest effect on LDs, iDGAT1 did exert a similarly small but significantly increase in tau uptake by KO cells. The graphs are presented at Fig. 6. The data suggest that DGAT1 is not primarily responsible for LD accumulation in RKO microglia, but that LD accumulation does have some effect on tau phagocytosis.

10) Include inflammation cytokines previously examined in Fig. 4a in the conditions used in Fig. 4e.

- We measured *Il1b* transcript in all groups and observed that iDGAT1 did not suppress TBE-mediated inflammation in either Cre- and Cre+ groups even if LDs were suppressed at that time as we saw in Fig. 6g and 6h. This data suggests that improved microglial tau phagocytic activity by iDGAT1 treatment is not mediated by suppression of inflammation, indicating that anti-inflammation can be a separate therapeutic approach. We discussed it at ‘Discussion’ part in our revised manuscript.

• **Figure 5**

11) Iba1 staining is mentioned in text for figure. 5c but representative images and analysis are not shown in the figure.

- As in response to reviewer 1 comments, this was an oversight. We updated the IBA1 data in Fig. 1d and 1e in the revised manuscript. We apologize.

12) Include astrocyte (GFAP) and microglia (Aif1) staining in LPS treated REV-ERB α microglia KO (Fig. 5e). Are changes between Cre- and Cre+ males significant?

- As we only have dissected brain tissues, but not the sectioned brains, from this experiment, we measured mRNA levels of *Gfap* and *Cd68* instead of staining. Both *Gfap* and *Cd68* levels were increased by LPS, but did not change by genotype, perhaps because the LPS effect overwhelmed the modest effect of REV-ERB α KO. The graphs are presented at Fig. S1e.

- LPS-injected male Cre+ mice have a trend toward increased IL1 β and IL6 in brain tissue as compared to male Cre- group. This effect may become significant with a larger N.

• **Figures 6 and 7**

13) The two figures are the same except male and female mice were analyzed. To better appreciate the sex-differences, the two figures should be combined and the displayed and quantified side-by-side (similar to Fig. 8).

- We combined male and female data at the same figure and presented at Fig. 2.

14) Higher resolution staining images (similar to Fig. 8b and c) need to be provided to properly evaluate the quality of the data.

- We updated the figure with high resolution images (Fig. 2b).

15) REV-ERB KO in microglia should be validated by co-staining for REV-ERB α with Iba1 or by qPCR of sorted microglia.

- We assessed REV-ERB α deletion using MACS CD45 bead-isolated microglia from tamoxifen (20mg/kg) treated *Cx3cr1::Cre^{ERT2}; Nr1d1^{fl/fl}* mice, and using 4-hydroxytamoxifen (1.5 μ M) treated *Cx3cr1::Cre^{ERT2}; Nr1d1^{fl/fl}* P0 pups. The deletion efficiency was ~50% *in vivo* and ~100% *in vitro*, respectively based on the mRNA levels of REV-ERB α (*Nr1d1*) compared to control Cre-. *In vivo* data was only presented in Fig. 1c based on the reviewer's comments.

Minor comments/suggestions:

16) There are a number of typos throughout the manuscript, e.g. “REV-ERB α causes induction of PLIN2 and LDs...” (line 1480149) should be REV-ERB α KO.

- Thank you, we have missed several of these errors and have attempted to correct them.

17) Remove representative flow cytometry plots to supplementary.

- We felt that the flow cytometry graphs in main figures are important for illustrating the variation in the data, so we left them in place for now. If we need to move all those graphs to supplementary figures for space reasons, we will do this process before final documentation.

18) It is puzzling that 18mon P301S tau mutant mouse brain was used to produce “Tau-enriched Brain Extract (TBE)” as these mice are known to develop hind limb paralysis by 9-12 months. Can authors comment on the phenotypes of the P301S mice used in their study?

- As the reviewer mentioned in here, we have seen hind limb paralysis as a phenotype of aged PS19 mice and some of them died before 18mon. However, there is substantial variability, and some live longer. When we sacrificed 18mon PS19 mice, they appeared poorly groomed and less active, less active, and close to death, but we did not perform behavior assays for measuring their motor dysfunction at that time.

Reviewer #3 (Remarks to the Author):

The key findings from this study suggest a strong sex-dependent link between lipid droplet (LD) accumulation in microglia and an upregulation of the housekeeping protein PLIN 2 on LD surfaces. The authors promote the idea that a circadian suppressor REV-ERBa deletion leads to microglia Plin2 accumulation in aged brain and that this is amplified by tau mediated inflammation. Activation of microglia with inflammogen LPS is strongly enhanced in male but not in female mice following REV-ERBa deletion. REV-ERBa deletion exacerbates pathology specifically in male P301S tau transgenic mice as assessed by IHC of neural cells and phagocytosis in microglia. Taken together the data suggest that REV-ERBa is a potential therapeutic target as a sex dependent regulator of microglial lipid metabolism in tauopathies.

These findings reported in the Communication are noteworthy, but data on lipid species and electrophysiology could make the claims stronger and open new avenues in therapeutic approaches in tauopathies. Missing the functional analyses (except phagocytosis) raises some concerns. In addition, concluding statement (abstract) that data demonstrate rather than suggest is a little exaggerated considering that no lipidomic analysis for neutral and other lipids was performed. However, findings are important and changes in microglia shown on increasingly complex models from in vitro to in vivo and limited but relevant human AD brain samples make these studies attractive.

1. The value of the work and its potential significance to the field is mainly in its logical combination of leads from numerous studies in the field which provided some stepping stones in biology of microglia and sex difference, lipid droplets and tau research. Thus, it is somewhat limited in novelty, but it is valuable as a study combining sex-dependent LD changes in microglia and the effects of circadian suppressor REV-ERBa on LD abundance and Plin2 expression. Sex dependence on LD changes is very convincing and highlights a greater susceptibility to tau pathology in males. A partial restoration of LD homeostasis by manipulation of REV-ERBa suggests a participation of other genes aside from REV-ERBa and these candidate genes are not indicated.

- We have now added RNA-seq data which shows remarkable metabolic changes in REV-ERBa KO microglia including changes in cell activation pathways and lipid metabolic processes. Since we focused on lipid metabolism, lipid-related genes that were altered by REV-ERBa KO are presented in Fig. 4c. This information shows widespread dysregulation of lipid metabolic genes in KO cells and may provide future candidate genes for further studies aspects of lipid regulation. We have also focused less on regulation of Plin2, which we know think is acting primarily as a marker of LDs, not a regulator.

2. The work provides relevant information from the current literature in the introduction, but lacks some of them in the discussion. Authors provide a link with earlier studies on LPS as an commonly used pro-inflammogen and compare it with uncharacterized cell culture media both of which produce sex-dependent effect. This part of the study is interesting but, the composition of this media is unknown and the main players are not suggested. New and important references on lipid droplet properties and composition should be included.

- Regarding weak points of discussion, we added diverse perspectives in ‘Discussion’ part on revised manuscript based on our additional experimental data.

- We assume that ‘uncharacterized cell culture media’ which was mentioned by reviewer 3 is a ‘TBE’. We have now performed both lipidomic and proteomic analysis using ‘TBE’ and the contents were normalized with dissolved control solution (Buffer H).

1) Proteomic results: Total 2,922 proteins were identified in TBE and ‘Microtubule-associated protein tau’ showed 52th abundance out of all the proteins, and was identified with 57% coverage of the protein. We did not mention and shared these data in manuscript, but we will present or upload it if the reviewer requested.

2) Lipidomic results; Total 174 features were classified and plotted for relative area. 1-oleoyl-sn-glycerco-3-phosphocholine was one of the major features in TBE. It also did not shared in manuscript like proteomic data.

- According to the lipidomic analysis, sphingolipids are sorted out as a dominant lipid class from TOP 20. We presented it at Fig. 7a and showed the structure of major sphingolipids (pSM) at Fig. 7e.

- We consider that these two-analysis including proteomic and lipidomic can be used for further publication in material-related journal. However, we could open these sources if the journal requested.

3. The work supports the major claims but there are some limitations. For instance, the only functional assay is phagocytosis of tau in microglia. Although it is an important assessment of microglia function it should be linked with neuronal functions by using electrophysiological techniques. Electrophysiological assessment of male and female REV-ERBa KO crossed with triple transgenic mice would be more convincing that the enhancement of LD abundance impairs neuronal functions. Cuervo et al., previously suggested a link between LD abundance in neurons and lipid transfer from neurons to glia. If the accumulation of LD is mainly in microglia due to the REV-ERBa KO but does not bear marked functional neuronal impairment the significance of the study is somewhat limited. Thus additional evidence for functional dependence of microglia LD on neuronal function is needed.

- We agree that this would be an interesting and exciting addition to the paper, but we feel that it is beyond the scope of what we can add to this particular manuscript. We do not have more mice available to generate live sections for electrophysiological experiments, and it would take at least a year to generate these. While we have not shown direct behavioral or electrophysiological endpoints, tau aggregation is very closely associated with neuronal damage and is universally considered to be an endpoint that is highly relevant to neurodegenerative disease.

4. Are there flaws in the data analysis or interpretation? There are no major flaws detected which would prohibit publication or require revisions. Additional experiments are needed:

(i) media used to show the induction of LD should be characterized (lipidomic and proteomic analysis is required to suggest at least few possible relevant players)

- The answers were described in question #2.

(ii) lipidomic analysis of LD should be performed.

- We greatly appreciate this suggestion and have now added collaborators to perform lipidomic analysis using VEH-treated and TBE-treated microglia. Interestingly, we sorted out dominantly expressed lipid feature, sphingolipid particularly palmitoyl sphingomyelin (pSM), by TBE treatment and observed that REV-ERB α KO microglia also showed upregulated sphingomyelin metabolic processes. The data are now presented in Fig. 7.

Table | List of TOP 20 lipids that expressed on TBE-treated microglia

Name	Formula	Molecular Weight	Reference Ion	Class	Microglia -VEH area	Microglia -TBE are	Log ₂ Fold change
SHexCer 42:2:2O	C48 H91 N O11 S	889.63128	[M-H] ⁻¹	Sphingolipids	4.21E+04	1.76E+07	8.71
Palmitoyl sphingomyelin	C39 H79 N2 O6 P	702.56757	[M+H] ⁺¹	Sphingolipids	5.27E+06	7.27E+08	7.11
PC(O-16:0/20:4)	C44 H82 N O7 P	767.58289	[M+H] ⁺¹	Glycerophospholipids	4.98E+06	1.59E+08	5.00
DG 14:0_22:6	C39 H64 O5	612.47537	[M+NH4] ⁺¹	Glycerolipids	1.51E+06	4.67E+07	4.95
Cer 18:2;20/24:1;(2OH)	C42 H79 N O4	661.60091	[M-H+HAc] ⁻¹	Sphingolipids	3.84E+04	7.87E+05	4.36
PC(14:0_16:1)	C38 H74 N O8 P	703.5152	[M+H] ⁺¹	Glycerophospholipids	2.61E+05	4.10E+06	3.97
PE O-16:1_20:4	C41 H74 N O7 P	723.52029	[M+H] ⁺¹	Glycerophospholipids	1.84E+06	2.38E+07	3.70
[M-CH3]-PC(16:0_14:0)	C38 H76 N O8 P	705.53085	[M+H] ⁺¹	Glycerophospholipids	8.69E+06	8.80E+07	3.34
DG 16:0_20:4	C39 H68 O5	616.50668	[M+NH4] ⁺¹	Glycerolipids	1.88E+06	1.52E+07	3.02
DG 18:1_22:5	C43 H72 O5	668.53798	[M+NH4] ⁺¹	Glycerolipids	5.37E+06	3.69E+07	2.78
Anandamide (AEA)	C22 H37 N O2	347.28243	[M+H] ⁺¹	Fatty Acyls	1.55E+05	1.05E+06	2.75
PC(17:0/18:1)	C43 H84 N O8 P	773.59345	[M+H] ⁺¹	Glycerophospholipids	5.72E+06	3.66E+07	2.68
PG(42:11)	C48 H73 O10 P	840.49413	[M-H] ⁻¹	Glycerophospholipids	2.50E+06	1.56E+07	2.64
Terpestacin	C25 H38 O4	402.27701	[M+H] ⁺¹	Organooxygen compounds	8.69E+05	4.93E+06	2.50
PC(16:1_16:1)	C40 H76 N O8 P	729.53085	[M+H] ⁺¹	Glycerophospholipids	2.18E+06	1.19E+07	2.45
PG(18:0/18:1)	C42 H81 O10 P	776.55673	[M-H] ⁻¹	Glycerophospholipids	5.16E+05	2.62E+06	2.34
DG 12:0_16:0	C31 H60 O5	512.44407	[M+NH4] ⁺¹	Glycerolipids	3.72E+06	1.73E+07	2.22
Docosanamide	C22 H45 N O	339.35012	[M+H] ⁺¹	Fatty Acyls	2.52E+06	1.11E+07	2.14
Lauric isopropanolamide	C15 H31 N O2	257.23548	[M+H] ⁺¹	Fatty Acyls	5.40E+06	2.33E+07	2.11
PG 44:12	C50 H75 O10 P	866.50978	[M-H] ⁻¹	Glycerophospholipids	9.07E+07	3.76E+08	2.05

(iii) electrophysiology to show the neuronal functions in REV-ERB KO should be conducted.

- Please see above. We do not have these capabilities in our lab, and would need to breed new cohorts of mice that would take too long to age to the point of characterization. Thus, we think that this is beyond the scope of the current study, but would be a very interesting follow-up experiment.

5. The methodology is sound, and the work meets the expected standards in the field. There is no novelty in this section.

- We understand our studies were not related to novel methodology. However, we believe that 'TBE' potentially can be a unique and useful material in AD field as a tau-mediated inflammatory stimuli instead of LPS. The new lipidomic and proteomic data also can be useful to future studies.

6. There is enough detail provided in the methods for the work to be reproduced but some justifications of selected times are needed.

For instance:

1. Why 7 days for phagocytosis of FITC-tau for EM and 2 hrs of FACS *in vitro*?

- The 7 day period was used to induce the fibrilization of tau *in vitro*. During this tau, tau is simply being shaken in buffer to allow aggregation prior to use. According to the protocol for fibrilization of tau peptide, 7 days were recommended, as EM data supported strong fibril formation that this time. The 2 hour timepoint is when we see optimal microglial engulfment of FITC-tau aggregates after microglial are treated with these aggregates acutely.

2. OA-induced LD formation can be partly reduced with pharmacological inhibitors of DGAT1 but this was not achieved with TBE suggesting that a complex “soup” of stimuli do not respond to these inhibitors and imply the contribution other than iDGAT. This is interesting but could be discussed in more detail if the composition of the TBE medium is analyzed.

-This is an important observation, and we agree that tis bears more study. Our proteomic assessment of TBE shows dozens of other proteins in the “soup”, and our data shows that TBE induces both LD formation and inflammatory responses. Our current RNA-seq data showed that cell activation is the most inducible biological process by REV-ERB α KO in microglia, suggesting that anti-inflammatory drugs could potentially recover defective microglial phagocytic activity with high efficiency. Perhaps, TBE caused severe inflammation and diverse metabolic changes that were hard to recover. These were discussed at ‘Discussion’ part in revised manuscript. It is also interesting that iDGAT does not fully block LD formation in REV-ERB α KO microglia, suggesting other pathways at work. The induction of palmitoyl sphingomyelin in TBE-treated microglia points to this pathway as possible contributor to LD formation and lipid dyshomeostasis.

3. Formation of LD by OA treatment is well established and serves here as one control but this is not a suitable control for TBE. At least some components of TBE must be identified and tested to show which of them induce LD formation and if these LD contain lipotoxins.

- We identified ‘sphingomyelin’ and observed that treatment of microglia with purified ‘sphingomyelin’ gradually increased LDs expression in microglia. The data was presented at Fig. 7f.

4. It is not only the appearance of LDs but also their quality

– i.e. lipid composition which will contribute to the functional impairment of neural cells. Thus, the following experiments are suggested:

1) determination of the major components in TBE which contribute to LD formation and their lipotoxicity

- The answers were described in question #2.

2) lipidomic analysis to determine the lipid species in microglia and neurons upon TBE stimulation.

- As we shown at Fig. 4, we determined and presented specific lipid features in microglia by TBE stimulation, but we didn't look in neurons. We assume the neuronal effects of TBE could be a great story and should be followed up upon after this work.

3) electrophysiological assessment of neuronal functions upon TBE stimulation in a sex-dependent manner.

- We apologized that we could not perform all electrophysiology-related experiment because of our limitations discussed above.

7. The strongest and most convincing evidence in the entire study is a sex dependent effects in REV-ERBa KO in the tau mouse model. It would be interesting to see if another circadian rhythm and clock gene is involved in lipid regulation in the investigated animal model. This aspect could be addressed in the discussion.

- This is a fascinating question! We have unpublished data showing that deletion of Bmal1, another clock gene upstream of REV-ERBa, can exert protective effects in the P301S mouse model, so the situation is highly dependent on cell type and specific clock gene.

REVIEWER COMMENTS

Reviewer #1 (Remarks to the Author):

I am satisfied with all revisions.

Reviewer #2 (Remarks to the Author):

In the original manuscript titled “Microglial REV-ERB α regulates lipid droplet formation and tauopathy in a sex dependent manner”, the authors attempted to convey the connection between microglial REV-ERB α with PLIN2 transcriptional regulation, lipid droplet accumulation, tau pathology and its sex dependency. While the authors provide additional data in the revised manuscript, these new results do not strengthen, but weaken these connections. Specifically, with regard to the regulation of PLIN2 and lipid droplets by REV-ERB α , the authors state in the rebuttal that “PLIN2 appears to be more of a marker of LDs than a regulator of them...Thus, we have de-emphasized PLIN2...”. As for sex dependency, which is a key selling point of the original manuscript, the authors now state “While we observe that this occurs primarily in male mice, we think this may be a product of the profound inflammatory difference between male and female microglia, and we do not pursue the underlying mechanisms of the sex difference in this paper.” As a result, both the novelty and significance of the revised manuscript are substantially reduced unfortunately. In addition, numerous suggested experiments were not performed due to technical limitations or lack of brain sections or animals.

Reviewer #3 (Remarks to the Author):

This is a revised version of the previously reviewed manuscript. The quality is considerably improved and the answers to the reviewer 3 are acceptable. The requested experiments (lipidomic and proteomic analyses) were performed but the data for the proteomic analysis are not provided in the supplemental section to the main text. They should be added.

There are still some limitations of the performed studies and those were not brought up. Please, summarize the major limitations of the study.

The key functional analyses i.e. electrophysiology was not performed and the reviewer is aware of their inability to perform such studies. As long as the authors indicate a lack of functional data for neurons, I would accept the current version of the manuscript after an additional editorial revision.

We appreciate the helpful criticism and have attempted to address these concerns in a thorough manner. The Reviewer remarks are written below in black, with our responses in **bolded red.**

Reviewer #1 (Remarks to the Author):

I am satisfied with all revisions.

We are glad to hear that and appreciate your consideration. Our revised manuscript was greatly improved by your recommendations.

Reviewer #2 (Remarks to the Author):

In the original manuscript titled “Microglial REV-ERB α regulates lipid droplet formation and tauopathy in a sex dependent manner”, the authors attempted to convey the connection between microglial REV-ERB α with PLIN2 transcriptional regulation, lipid droplet accumulation, tau pathology and its sex dependency. While the authors provide additional data in the revised manuscript, these new results do not strengthen, but weaken these connections. Specifically, with regard to the regulation of PLIN2 and lipid droplets by REV-ERB α , the authors state in the rebuttal that “PLIN2 appears to be more of a marker of LDs than a regulator of them... Thus, we have de-emphasized PLIN2...”. As for sex dependency, which is a key selling point of the original manuscript, the authors now state “While we observe that this occurs primarily in male mice, we think this may be a product of the profound inflammatory difference between male and female microglia, and we do not pursue the underlying mechanisms of the sex difference in this paper.” As a result, both the novelty and significance of the revised manuscript are substantially reduced unfortunately. In addition, numerous suggested experiments were not performed due to technical limitations or lack of brain sections or animals.

We understand this concern, as we have changed the mechanistic focus of the paper considerably after completing many of the recommended experiments. However, we feel that the paper still has considerable novelty by showing that microglial REV-ERB α regulates lipid metabolism and tau pathology in a sex dependent manner. Our newer findings suggest that regulation of PLIN2 does not explain the entire effect, and that multiple lipid and inflammatory pathways are altered in microglia after REV-ERB α deletion. Our new data shows that, under basal conditions in culture, preventing the induction of Plin2 expression in microglia does not prevent LD formation after REV-ERB α deletion. Thus, PLIN2 regulation by REV-ERB α may be part of the story, or could be a result of increased LD formation due to alternative pathways. Our new data expands the scope of the lipid metabolic changes regulated by REV-ERB α in microglia, rather than suggesting that PLIN2 regulation mediates all of it. Our finding that REV-ERB α represses Plin2 expression at the transcription level is still important, as many researchers in lipid metabolism field have used Plin2 as a representative marker of LDs, but this relationship with REV-ERB α was unknown. Our analysis and discussion of sex-dependent differences in the effect of microglial REV-ERB α deletion has not changed in the revision, as we still show that the effects of REV-ERB α on tau-induced LD formation, LPS responses, and tauopathy are different between males and females. We feel that this is still an important and informative aspect of the manuscript, though uncovering this mechanism is perhaps too tall an order for this manuscript. It has been known for years that females have a higher risk of AD, and that male microglia have different inflammatory characteristics, yet a firm mechanistic understand of either phenomena is lacking, despite dozens of papers on the subject. Thus, by revealing this sex-dependent effect of REV-ERB α and strongly characterizing it, we can set the stage for future studies to understand in further. We have now mentioned this limitation in the Discussion.

Reviewer #3 (Remarks to the Author):

This is a revised version of the previously reviewed manuscript. The quality is considerably improved and the answers to the reviewer 3 are acceptable. The requested experiments (lipidomic and proteomic analyses) were performed but the data for the proteomic analysis are not provided in the supplemental section to the main text. They should be added. There are still some limitations of the performed studies and those were not brought up. Please, summarize the major limitations of the study. The key functional analyses i.e. electrophysiology was not performed and the reviewer is aware of their inability to perform such studies. As long as the authors indicate a lack of functional data for neurons, I would accept the current version of the manuscript after an additional editorial revision.

We appreciate your time and efforts examining the revised paper. As you requested, we have now updated proteomics data in Supplementary Figure, Methods, and Results. A table with the proteomics data is now provided in Supplemental Fig. S4. Raw data has also been uploaded through the FigShare website.

We appreciate your understanding regarding function/electrophysiology tests. We have added a paragraph in the Discussion which highlights these limitations of our study, along with several others.

“Our study has several additional limitations. We were not able to examine behavioral changes related to the pathological changes seen in our mice. The effects of microglial REV-ERB α deletion on neuronal function, in terms of synaptic plasticity and electrophysiology, were also not examined, and remain unknown. However, the increased tau aggregation and neuronal loss in CA1 that we have shown are indicative of neuronal damage and are associated closely with neuronal dysfunction and cognitive impairment in mice and humans¹⁹. The specific molecular underpinnings of the sex differences observed are also not yet understood, though the same can be said for sex differences in AD in general.”

While it does not address neuronal function, we have now added some data showing increased loss of neurons in the CA1 region of microglial REV-ERB α KO;PS19 mice (NeuN staining and quantification). This at least suggests damage to neurons, which would be expected to negatively impact function.